# Form, function, and divergence of a generic fin shape in small cetaceans

Vadim Pavlov [1,2]*, Cecile Vincent[3], Bjarni Mikkelsen[4], Justine Lebeau[5], Vincent Ridoux[3], Ursula Siebert [2]

**1** Hopkins Marine Station, Stanford University, Pacific Grove, CA, United States of America, **2** Institute for Terrestrial and Aquatic Wildlife Research, The University of Veterinary Medicine Hannover, Foundation, Buesum, Germany, **3** Centre d'Études Biologiques de Chizé, Université de La Rochelle, La Rochelle, France, **4** Havstovan/Faroe Marine Research Institute, Tórshavn, Faroe Islands, **5** Scripps Research Institute, La Jolla, CA, United States of America

\* pavlov.v.v@gmail.com

**Data Availability Statement:** All relevant data are within the manuscript and its Supporting Information files.

## Abstract

Tail flukes as well as the dorsal fin are the apomorphic traits of cetaceans which appeared during the evolutionary process of adaptation to the aquatic life. Both appendages present a wing-like shape associated with lift generation and low drag. We hypothesized that the evolution of fins as lifting structures led to a generic wing design, where the dimensionless parameters of the fin cross-sections are invariant with respect to the body length and taxonomy of small cetaceans (Hypothesis I). We also hypothesized that constraints on variability of a generic fin shape are associated with the primary function of the fin as a fixed or flapping hydrofoil (Hypothesis II). To verify these hypotheses, we examined how the variation in the fin's morphological traits is linked to the primary function, species and body length. Hydrodynamic characteristics of the fin cross-sections were examined with the CFD software and compared with similar engineered airfoils. Generic wing design of both fins was found in a wing-like planform and a streamlined cross-sectional geometry optimized for lift generation. Divergence in a generic fin shape both on the planform and cross-sectional level was found to be related with the fin specialization in fixed or flapping hydrofoil function. Cross-sections of the dorsal fin were found to be optimized for the narrow range of small angles of attack. Cross-sections of tail flukes were found to be more stable for higher angles of attack and had gradual stall characteristics. The obtained results provide an insight into the divergent evolutionary pathways of a generic wing-like shape of the fins of cetaceans under specific demands of thrust production, swimming stability and turning control.

## Introduction

The question of the role of dolphin appendages as lift-generating surfaces is related to the evolutionary process of adaptation of marine mammals to the life in a moving fluid. In this context, the dorsal fin and tail flukes of cetaceans are of particular interest, as there is no evidence of their analogs in terrestrial ancestors [1–3], and their appearance in cetaceans is presumably associated with transition from drag-based to lift-based locomotion in an aquatic environment [4,5]. As a *de novo* dermal structure [6], the dorsal fin and tail flukes can be described with a

**Funding:** The work of Vadim Pavlov was funded by the German Science Foundation (SI 1542/1).

**Competing interests:** The authors have declared that no competing interests exist.

limited set of morphological traits, where the relation between the traits and wing performance can be unambiguously interpreted. This unambiguous interpretation provides insight into the evolutionary pathways to divergence of a generic shape driven by the different demands in stabilizing the straight-line swimming, turning control and thrust production.

Both appendages represent an underwater wing, where the fin span ($S$) and fin planform area ($A$) correlate with the body length ($BL$) of cetaceans [3,7,8]. The relationship between $S$, $A$, and $BL$ is different through the life history stages [7,8], this possibly being associated with the different patterns of swimming in calves and adult animals [9].

The planform of tail flukes most often presents a falcate, swept-back tapered outline, with moderate or high aspect ratio $AR = S^2/A$, ranging from 2.0 for the Amazon river dolphin *Inia geoffrensis* to 6.2 for the false killer whale *Pseudorca crassidens* [3,7,8]. The dorsal fins of the different cetacean species normally have lower $AR$ and more variable planform with positive, neutral and negative sweep of the trailing edge that appears as falcate-shaped, rounded and triangular-shaped fins [10]. The cross-sectional design of both dorsal fins and flukes displays a symmetrical streamlined outline with a rounded leading edge [11–16]. This shape is comparable with the engineered airfoils and hydrofoils [11,15–17].

The combination of the moderate aspect ratio, sweep, cross-sectional design and flexibility of the fins characterizes the efficient underwater wing [18–21]. Meanwhile, there is a fundamental difference in the operational mode of the dorsal fin as a fixed wing and the tail flukes acting as a pair of flapping wings [11,18]. Apart from the fixed wing, a flapping wing is involved in specific mechanisms of lift and drag generation dealing with leading edge vortex and wake capture [22]. The advantage of the flapping mode is low drag, increased lift, delayed stall and a wider range of the angles of attack [23,24].

In this study, we hypothesize that the evolution of fins as lifting structures led to a generic wing design, where the dimensionless parameters of the fin cross-sections are invariant with respect to the body length and taxonomy of small cetaceans **(Hypothesis I)**. We also hypothesize that constraints on variability of a generic fin shape are associated with the primary function of the fin as a fixed or flapping hydrofoil **(Hypothesis II)**. To verify these hypotheses, we examined how the variation in the fin's morphological traits is linked to the primary function, species and body length.

This study focuses on the analysis of 2D sections of the fin and their span-wise variation to gain insight into optimization a generic fin shape regarding the primary function. Our approach was to compare the dolphin fins with their engineered analogs performing a similar function and optimized for a certain range of the operational conditions. The shape and hydrodynamic performance of the fin cross-sections was compared with the engineered foils in terms of standard airfoil parameters, lift *(Cl)*, drag *(Cd)* and moment *(Cm)* coefficients. We assumed that cross-sectional design of the dorsal fin as a vertical stabilizer could be optimized for the narrow range of small angles of attack. Furthermore, we assumed that the cross-sections of the tail flukes, as a flapping propulsor, could be optimized for the wider range of the angles of attack. To check these assumptions, we compared the dorsal fin and tail flukes' cross-sections with the hydrofoils and airfoils used for the yacht keels and rudders, and the aerobatic wings, respectively. The obtained results could serve as a starting point for further studies on the effect of span-wise and chord-wise bending of the bio-inspired flapping foils in wake formation and thrust generation.

## Materials and methods

### Sampling

Measurements of the body length and fins were taken from representatives of five genera of the family *Delphinidae* and one genus of the family *Phocoenidae*, these having different body length,

external morphology and specialization (Fig 1). Dorsal fins and tail flukes in good condition were taken from dead stranded and by-caught animals in the Bay of Biscay, Black Sea, North Sea, and the Norwegian Sea. Information about stranded and by-caught animals was obtained from the national and local stranding networks, and individuals. Most of bycatch had been caught by commercial fishermen using the gill nets and trawlers in the Bay of Biscay and Black Sea. The authorization for collecting the specimens was obtained from the PELAGIS UMS 3462 La Rochelle University/CNRS, Ministry of the Environment of Ukraine, Ministry of Energy, Agriculture, Environment, Nature and Digitalization of Schleswig-Holstein, and Faroese Museum of Natural History conducting marine mammal research under a special permit issued by the Faroese Government. The IUCN status of species studied is presented in S1 File.

Dorsal fin measurements were taken from three harbor porpoises *Phocoena phocoena*, sixteen common dolphins *Delphinus delphis*, ten bottlenose dolphins *Tursiops truncatus*, eleven Atlantic white-sided dolphins *Lagenorhynchus acutus*, three Atlantic white-beaked dolphins *Lagenorhynchus albirostris* and thirteen long-finned pilot whales *Globicephala melas*. Due to the limited number of the tail flukes in good condition, measurements of the flukes were taken from three *P. phocoena*, four *D. delphis*, three *T. Truncatus*, three *L. acutus*, three *L. albirostris*

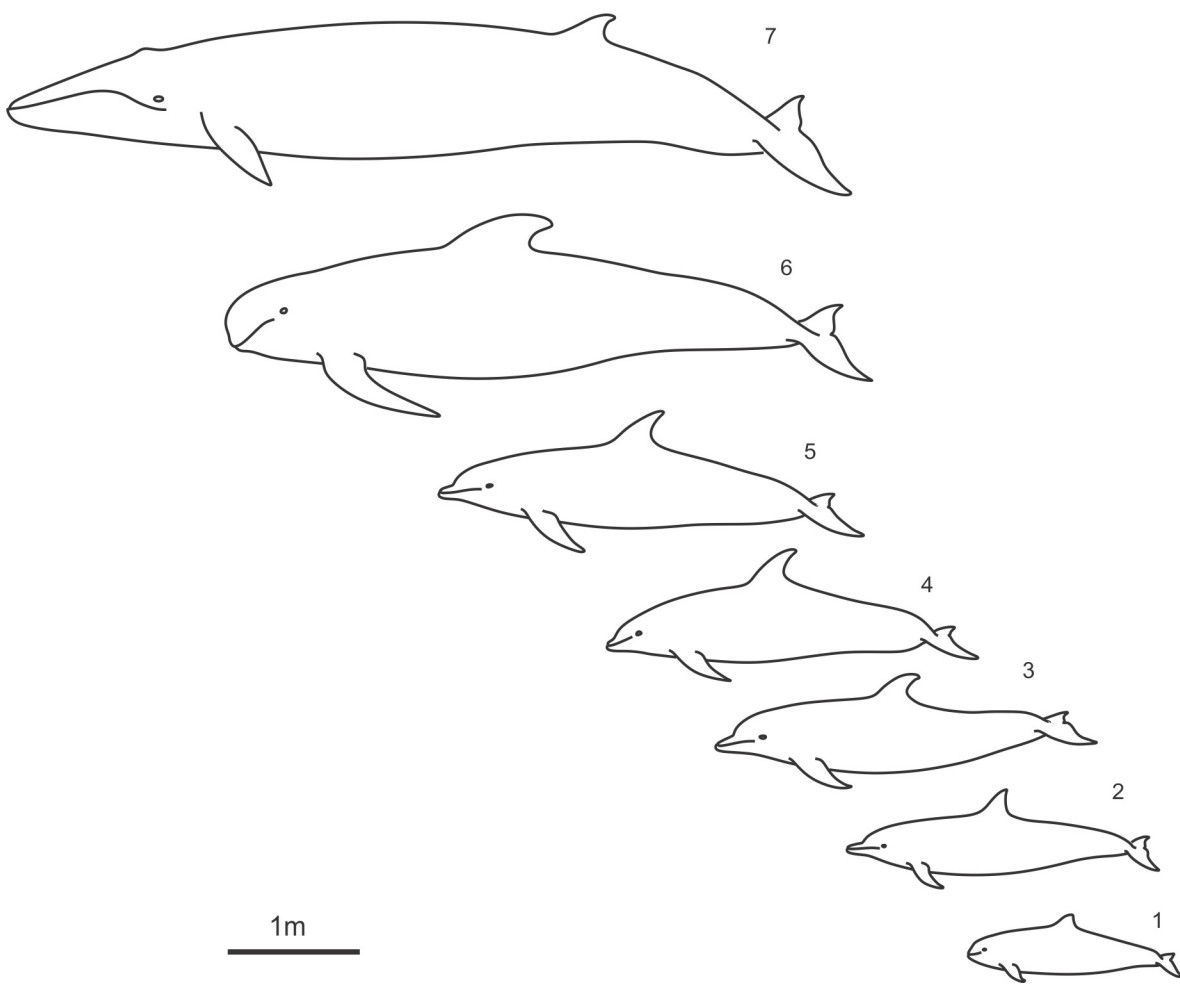

**Fig 1. Small cetacean species selected for this study.** 1 –*Phocoena phocoena*, 2 –*Delphinus delphis*, 3 –*Tursiops truncatus*, 4 –*Lagenorhynchus acutus*, 5 –*Lagenorhynchus albirostris*, 6 –*Globicephala melas*, 7 –*Balaenoptera acutorostrata*.

and three *G. melas*. Measurements were taken on one fluke, left or right, depending on the condition. Additionally, the measurements of the dorsal fin and flukes were taken from one specimen of the Minke whale *Balaenoptera acutorostrata* (Fig 2).

Fin span was measured from tip to tip in tail flukes and from the root chord to the top on the dorsal fin. The position of root chord was assumed to be a line parallel to the long axis of the body passing through the point of the maximal curvature on the leading edge at the base of the fin [15]. Fins were cut off from the body and six cross-sections parallel to the long axis of the body were made at equal intervals. Photographs of the intact fins and cross-sections were taken with the ruler as a scale. In total, 462 cross-sections were processed, measured and analyzed, 342 for the dorsal fin and 120 for the tail flukes.

## Outline extraction

Photographs of the fin planform and cross-sections were imported into the AutoCAD software and calibrated using ruler markings. The fin planform outline and cross-sections were drawn manually using the cubic B-spline tool in AutoCAD. As the fin cross-sections usually perform certain bends, the linear approximation procedure was applied to straighten the outline. The outline was divided into two parts using extreme points of the leading and trailing edges. A total of 100 points were placed on each part at equal intervals. Each pair of opposite points was joined by the complementary segment. Then the middle line passing through the middle points of all complementary segments was drawn. Coordinates of the middle and end points of each segment as well as the length of the middle line were used to calculate the straightened outline's coordinates.

## Wing and airfoil parameters of the fins

The following wing parameters were measured and calculated on the images of fins (Fig 3):

1. Fin span *S* in cm, measured from the fin base to the fin tip.

2. Basal length *BL* of the fin in cm, measured as length of the line parallel to the long axis of the body and starting from the point of maximal curvature on the leading edge.

3. Leading edge length *L* in cm, measured from point of maximal curvature on the leading edge to the fin tip.

4. Fin area *A*, in cm$^2$, calculated with projection of the fin on the plane.

5. Angle of sweep Λ in degrees, measured as angle between a perpendicular to root chord at the base of the fin and one-quarter chord position (Fig 4).

6. Aspect ratio *AR* calculated as $S^2/A$.

7. Canting index *CI* calculated by the formula [7]:

$$CI = \frac{(L^2 - H^2)^{1/2}}{BL}$$

where *L* = leading edge length, *H* = span of the fin, and *BL* = basal length of the fin.

On straightened outlines of the cross-sections, the following airfoils parameters were measured and calculated (Fig 5):

1. Chord length *CL* in mm.

2. Maximal thickness *MT* in mm.

3. Position of maximal thickness *PMT* measured from the leading edge in mm.

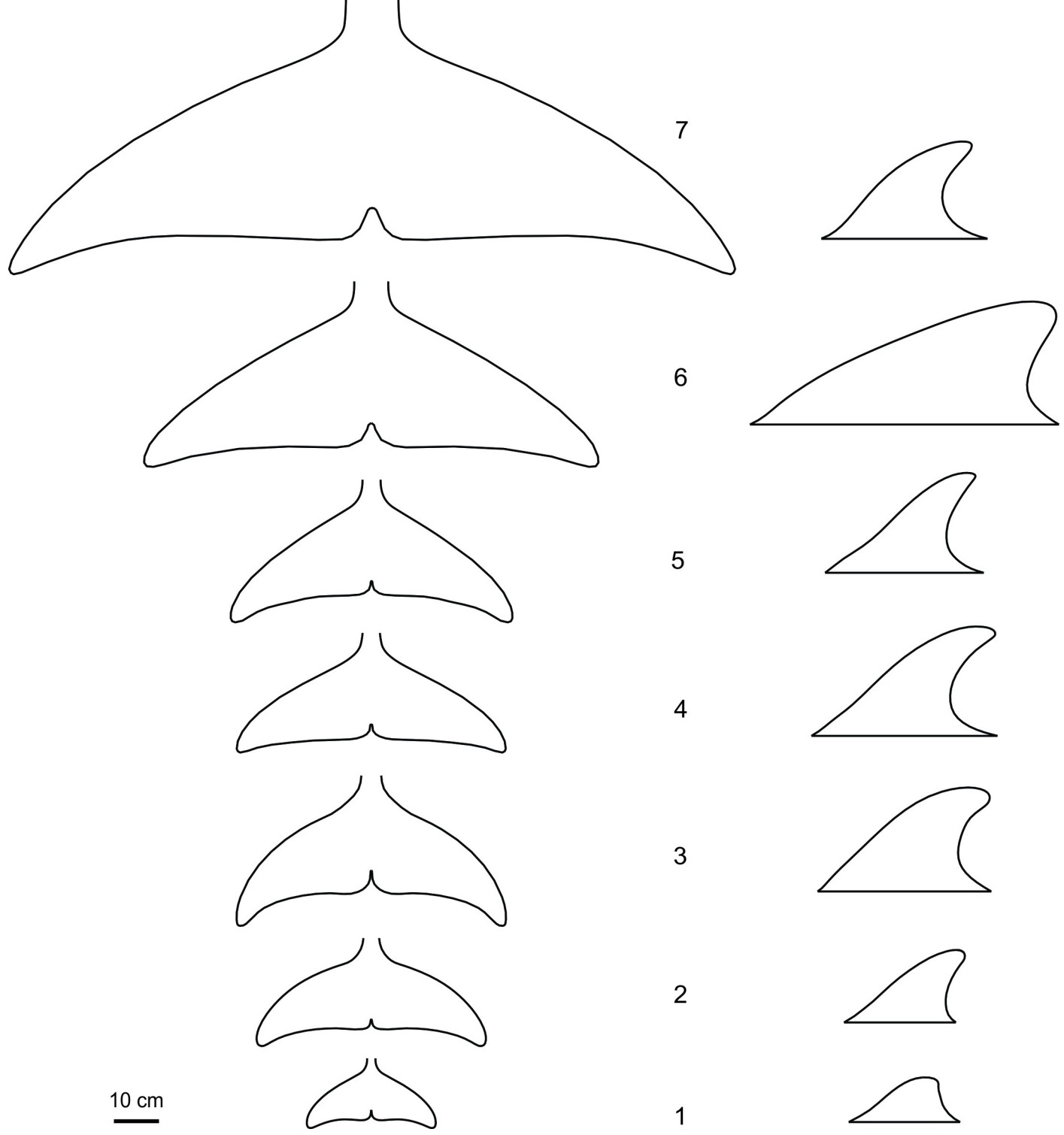

10 cm

**Fig 2. Dorsal fin and tail fluke's outline.** 1 –*Phocoena phocoena*, 2 –*Delphinus delphis*, 3 –*Tursiops truncatus*, 4 –*Lagenorhynchus acutus*, 5 –*Lagenorhynchus albirostris*, 6 –*Globicephala melas*, 7 –*Balaenoptera acutorostrata*.

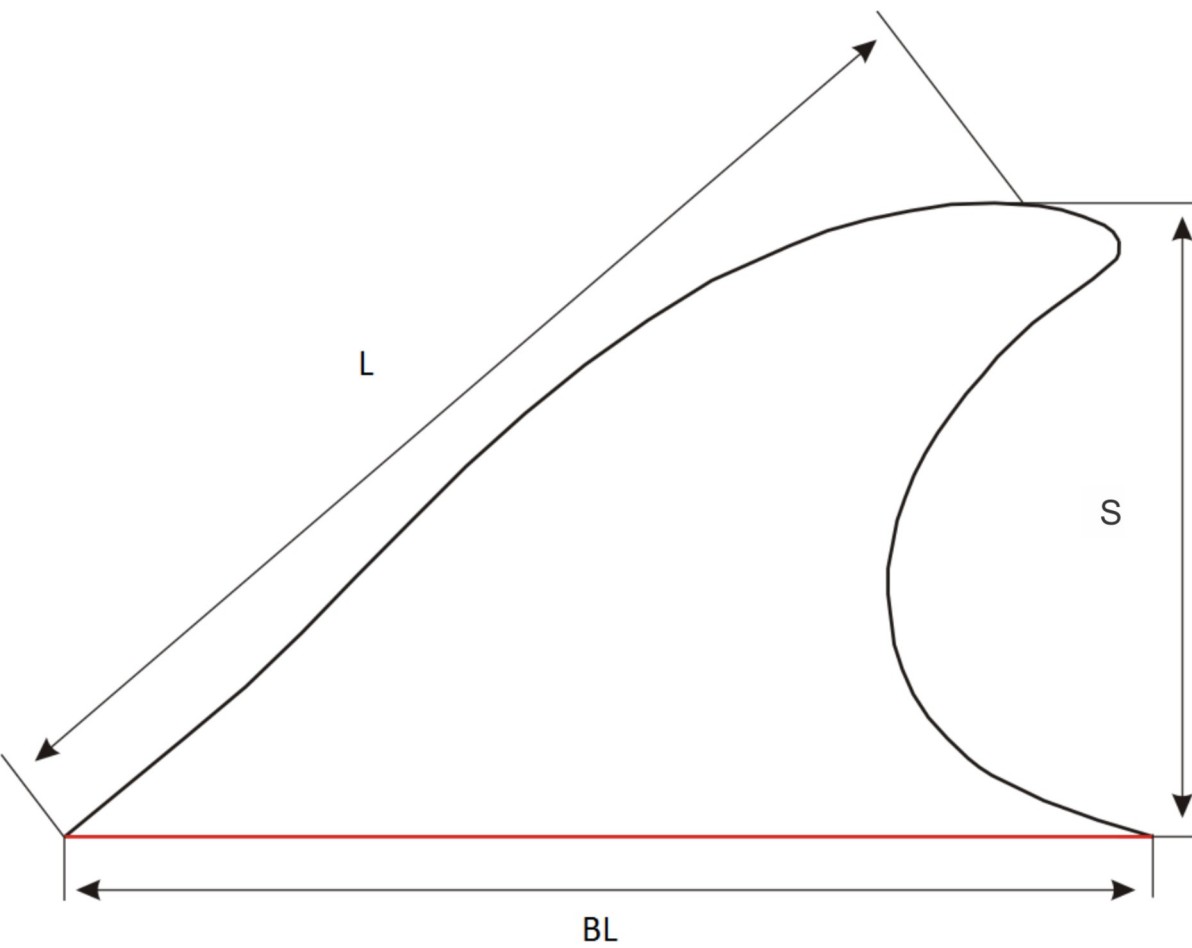

**Fig 3. Basic measurements of fins.**

4. Leading edge radius *LER* in mm.

## Hydrodynamic characteristics of the cross-sections of fins

Cross-sections of fins were analyzed with the DesignFOIL™ computer fluid dynamics software (DreeseCode Software). The water flow around the cross-sections and similar airfoils was simulated with a panel method. DesignFOIL™ software breaks the airfoil into many panels and forces the velocity at each panel to be tangential to that surface. Conglomerating all of these velocities leads to the velocity distribution and therefore the pressure coefficient distribution. The laminar flow portion of the boundary layer solver was based on the approximation of the Karman and Pohlhauson method [17]. The turbulent flow was modeled on the approximation Buri method [25]. The results of the validation of the DesignFOIL™ software using the wind tunnel data can be found here [26].

The experimental design included the utilization of the chord-normalized coordinates of the cross-sections and two selected speeds, 2 m/sec and 8 m/sec, to study the hydrodynamic performance of the cross-sections in terms of lift coefficient $Cl$, drag coefficient $Cd$, moment coefficient $Cm$ and pressure coefficient $Cp$. Cetacean species present a wide range of swimming speeds that can be arbitrarily divided into a sustained speed of swimming, where an activity level is maintained for hours, and fast speed of swimming, where an extreme activity

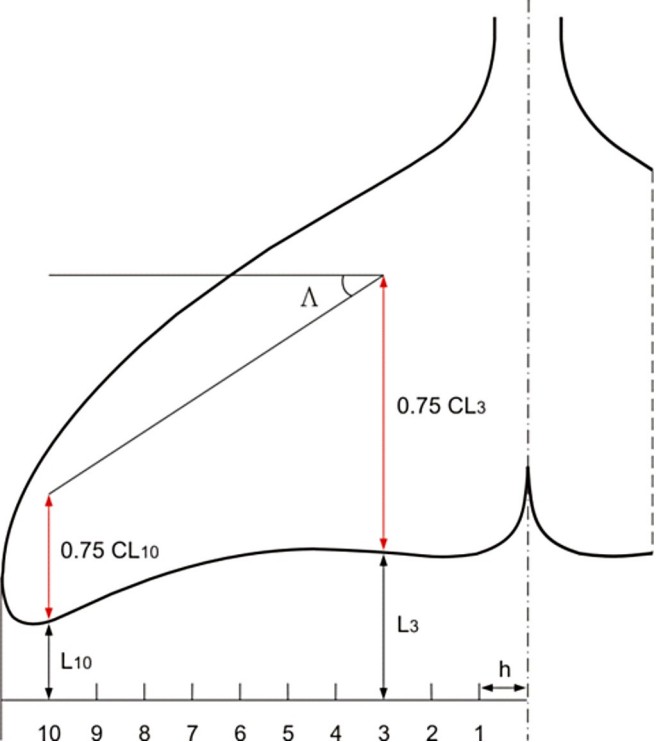

**Fig 4. Scheme of measurement of sweep angle Λ on fins.**

level is maintained for seconds. The first range embraces the variety of swimming behaviors including routine activities, cruising and migrating. The second range is associated with chasing the prey and escaping the predators [27]. The optimal speed of swimming minimizing the cost of transport and calculated based on the empirical allometric relationships between swim speed and body mass for marine mammals (speed = $0.78\text{mass}^{0.10}$) [28] varied from 1.2 m/sec for *P. phocoena* to 1.8 m/sec for *B. acutorostrata* (Table 1). The observed swimming speeds of species selected for our study varied in range from 0.5–4.2 m/sec for the sustained speed of swimming and 4.6–8.3 m/sec for the fast speed of swimming (Table 1). In the absence of published data on the observed maximum speed of *G. melas* and *L. acutus*, we assumed that this

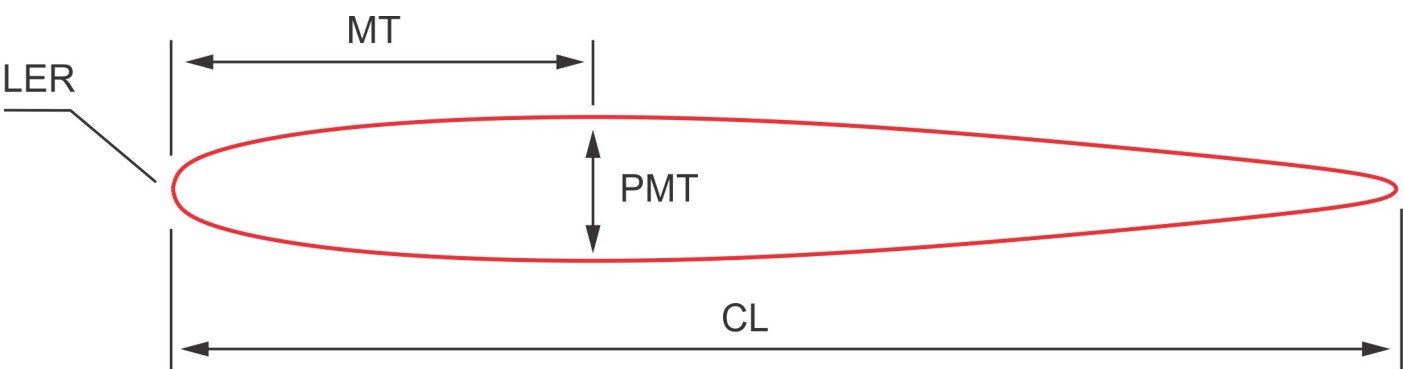

**Fig 5. Scheme of the airfoil parameters measured on the cross-sections of fins.** *CL*–chord length, *MT*–maximal thickness, *PMT*–position of maximal thickness, *LER*–leading edge radius.

**Table 1. Predicted and observed speeds of swimming for the selected species.**

| Species | Mass, kg | Predicted optimal speed, m/sec | Observed sustained speed, m/sec | Observed maximum speed, m/sec | Source |
|---|---|---|---|---|---|
| *G. melas* | 731 | 1.5 | 0.5–4.2 | N/A | [30,31] |
| *L. acutus* | 136 | 1.3 | 1.4–3.95 | N/A | [32,33] |
| *T. truncatus* | 245 | 1.4 | 1.7–4.2 | 6–8.2 | [34–36] |
| *D. delphis* | 115 | 1.3 | 1.6–2.8 | 8 | [36–38] |
| *L. albirostris* | 234 | 1.3 | 1.6–3.3 | 8.3 | [39,40] |
| *P. phocoena* | 74 | 1.2 | 0.5–1.9 | 4.6–6.2 | [41–43] |
| *B. acutorostrata* | 4500 | 1.8 | 0.5–3.3 | 7.2 | [44,45] |

may be comparable with the closely related species, namely 9 m/sec for the short-finned pilot whale *G. macrorhynchus* [27] and 7.7 m/sec for the Pacific white-sided dolphin *L. obliquidens* [29]. Two speeds, 2 m/sec and 8 m/sec, chosen for our CFD testing of the fin cross-sections fell within a range of observed sustained and fast swimming speeds, respectively.

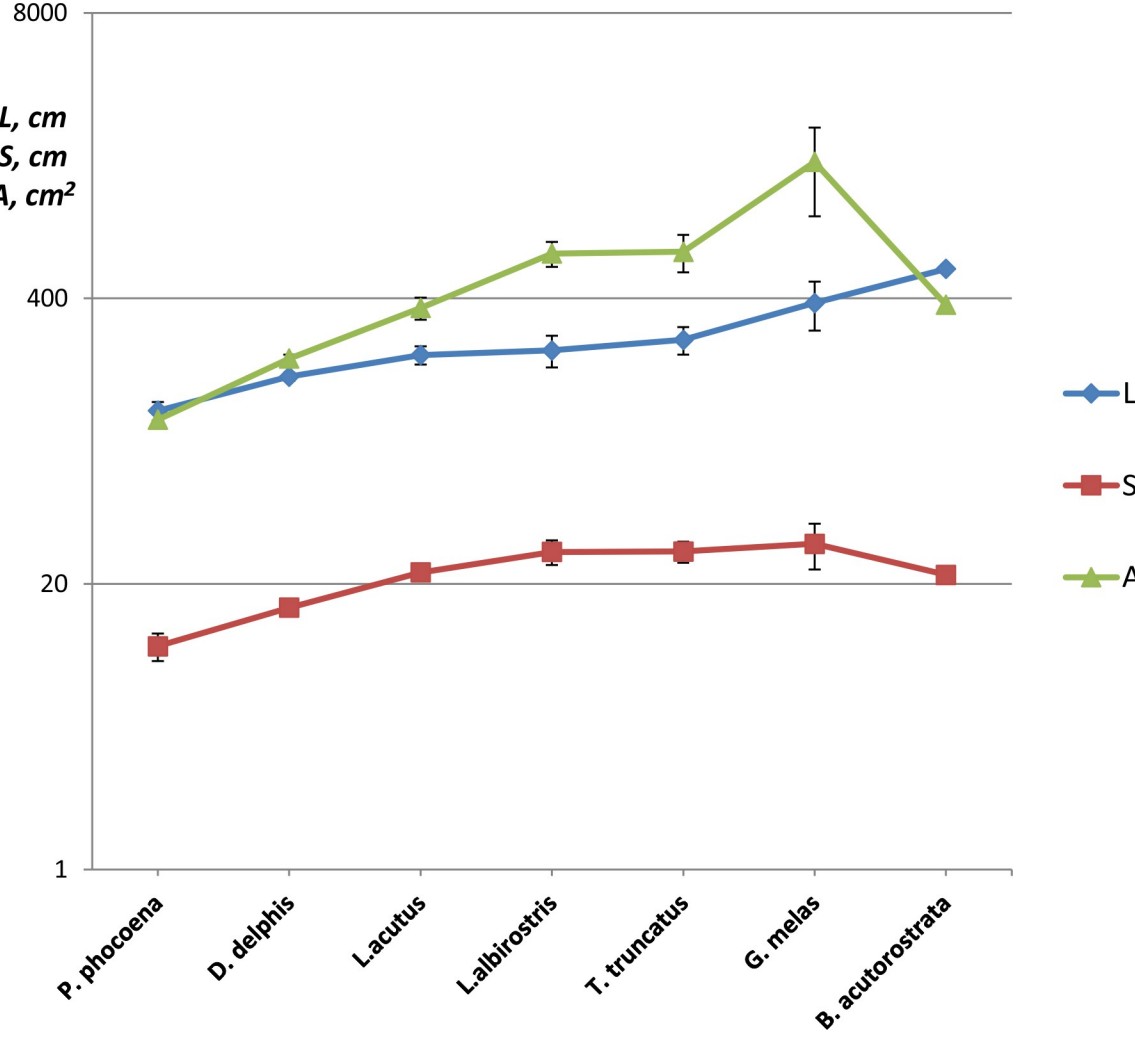

**Fig 6. Species-specific differences in the body length *L* cm, span of the dorsal fin *S* cm and area of the dorsal fin *A* cm², means ± SD.**

**Table 2. Dimensional and dimensionless parameters of the dorsal fins, means ± SD.**

| Species | BL | S, cm | A, cm2 | Λ, degs | AR | CI |
|---|---|---|---|---|---|---|
| *P. phocoena* | 123 ±12 | 10 ± 1.5 | 112 ± 8 | 40 ± 4 | 0.77 ± 0.07 | 0.31 ± 0.10 |
| *D. delphis* | 176 ± 7 | 15.6 ± 1 | 213 ± 9 | 48 ± 3 | 1.15 ± 0.14 | 0.30 ± 0.04 |
| *L. acutus* | 221 ± 21 | 22.5 ± 1.4 | 362 ± 42 | 44 ± 3 | 1.41 ± 0.13 | 0.22 ± 0.06 |
| *L. albirostris* | 232 ± 38 | 28 ± 3.6 | 640 ± 83 | 48 ± 2 | 1.27 ± 0.12 | 0.47 ± 0.06 |
| *T. truncatus* | 259 ± 37 | 28.1 ± 3.1 | 652 ± 126 | 47 ± 3 | 1.22 ± 0.16 | 0.30 ± 0.03 |
| *G. melas* | 381 ± 98 | 30.4 ± 7.1 | 1674 ± 728 | 63 ± 2 | 0.58 ± 0.05 | 0.32 ± 0.06 |
| *B. acutorostrata* | 545 | 22 | 374 | 45 | 1.29 | 0.14 |

Species are ordered according to the body length BL.

Hydrodynamic characteristics of fin cross-sections and the conventional airfoils having a similar outline was compared in terms of *Cd* and *Cp*. For all cross-sections, a comparison was made at zero angle of attack α, formed by the chord of a cross-section and the direction of the flow. For the cross-sections taken at the base and the top of the fins, and for the corresponding airfoils, the *Cd*, *Cl* and *Cm* were calculated for the range of α from 0 to 20° and plotted in a drag polar diagram.

For comparison purposes, span-wise lift distribution on the cross-sections of the dorsal fin and fluke of the *D. delphis* was calculated as $CL^*Cl_{max}$, where *CL* in mm is a chord length of a symmetrical cross-section, and $Cl_{max}$ is a maximal lift coefficient.

### Statistical analysis

ANOVA was performed to examine the relation between the dimensionless airfoil parameters *MT%CL*, *PMT%C*, and *LER%CL* of the fin cross-sections and fin type (dorsal fin or flukes), species, position of the cross-section on the fin (Section #) and body length. The fin type factor of the ANOVA had two levels (dorsal fin, tail flukes), the species factor had six levels (*P. phocoena*, *D. delphis*, *T. truncatus*, *L. acutus*, *L. albirostris*, *G. melas)*, and the position on the fin factor had six levels (Section #1 –Section #6). Principal Component Analysis (PCA) was performed to describe the patterns of cross-sectional geometry variation in both types of fins.

## Results

### Shape and cross-sectional geometry of the dorsal fins

Both size and shape of the dorsal fins varied significantly among the studied species. Fin span varied from S = 10 ± 1.5 (means ± SD) cm in *P. phocoena* to S = 30.4 ± 7.1 (means ± SD) cm in

**Table 3. Correlation matrix of the dimensional and dimensionless parameters of the dorsal fins in the Odontoceti species.**

| Variables | BL | S, cm | A, cm² | Λ, rad | AR | CI |
|---|---|---|---|---|---|---|
| **BL** | **1** | **0.873** | **0.967** | **0.916** | -0.318 | 0.078 |
| **S, cm** | **0.873** | **1** | 0.779 | 0.674 | 0.089 | 0.296 |
| **A, cm²** | **0.967** | 0.779 | **1** | **0.945** | -0.524 | 0.179 |
| **Λ, rad** | **0.916** | 0.674 | **0.945** | **1** | -0.515 | 0.157 |
| **AR** | -0.318 | 0.089 | -0.524 | -0.515 | **1** | -0.043 |
| **CI** | 0.078 | 0.296 | 0.179 | 0.157 | -0.043 | **1** |

Values in bold are different from 0 with a significance level alpha = 0.05.

*G. melas*. Span *S* and area *A* of the fin increased with increasing length of the body in the species (Fig 6). Distinctions in the fin shape were revealed in dimensionless parameters *AR* and

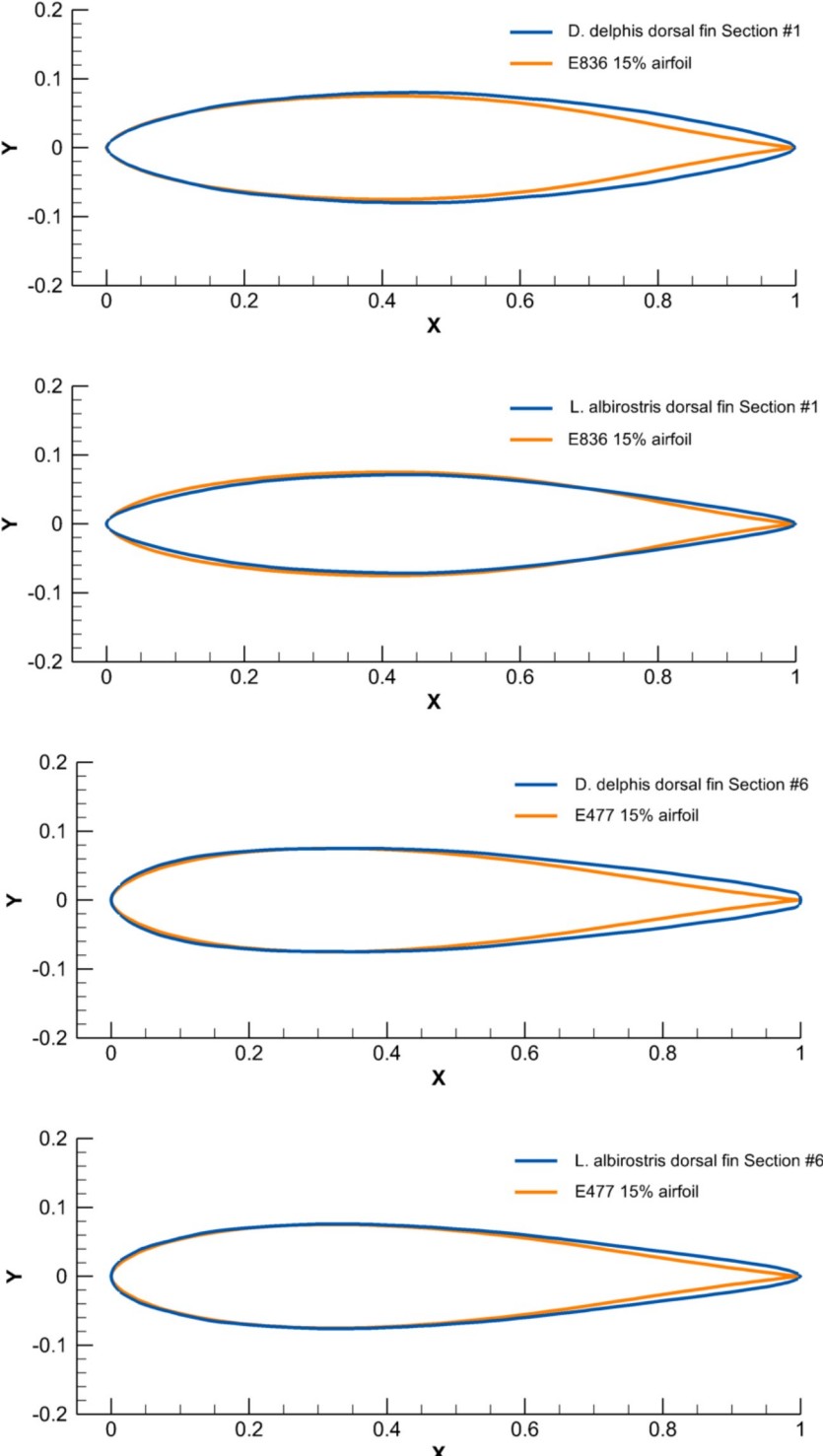

**Fig 7. Comparison of the chord-normalized profile coordinates of cross-sections taken at the base and top of the dorsal fin of the selected species with the conventional airfoils.**

**Table 4. Dimensional and dimensionless parameters of the tail flukes, means ± SD.**

| Species | BL | S, cm | A, cm2 | Λ, degs | AR |
|---|---|---|---|---|---|
| *P. phocoena* | 123 ±12 | 30 ± 3 | 220 ± 41 | 32 ± 1 | 4 ± 0.1 |
| *D. delphis* | 176 ± 7 | 52 ± 4 | 599 ± 98 | 29 ± 2 | 4.5 ± 0.1 |
| *L. acutus* | 221 ± 21 | 63 ± 7 | 879 ± 262 | 34 ± 3 | 4.6 ± 0.3 |
| *L. albirostris* | 232 ± 38 | 61 ± 11 | 835 ± 266 | 32 ± 1 | 4.5 ± 0.2 |
| *T. truncatus* | 259 ± 37 | 61 ± 5 | 865 ± 126 | 35 ± 1 | 4.3 ± 0.1 |
| *G. melas* | 381 ± 98 | 103 ± 12 | 1947 ± 377 | 28 ± 2 | 5.5 ± 0.2 |
| *B. acutorostrata* | 545 | 165 | 4614 | 27 | 5.9 |

Species are ordered according to the body length BL.

*CI* (Table 2) as well as in the sweep of the trailing edge of the fin. The fin shape varied from a triangular one with low *AR* and positive sweep of the trailing edge in *P. phocoena* to a falcate-shaped one with moderate or high *AR* and negative sweep of the trailing edge in *D. delphis*, *T. truncatus*, *L. acutus*, *L. albirostris* and *B. acutorostrata*. Within the latter group, the

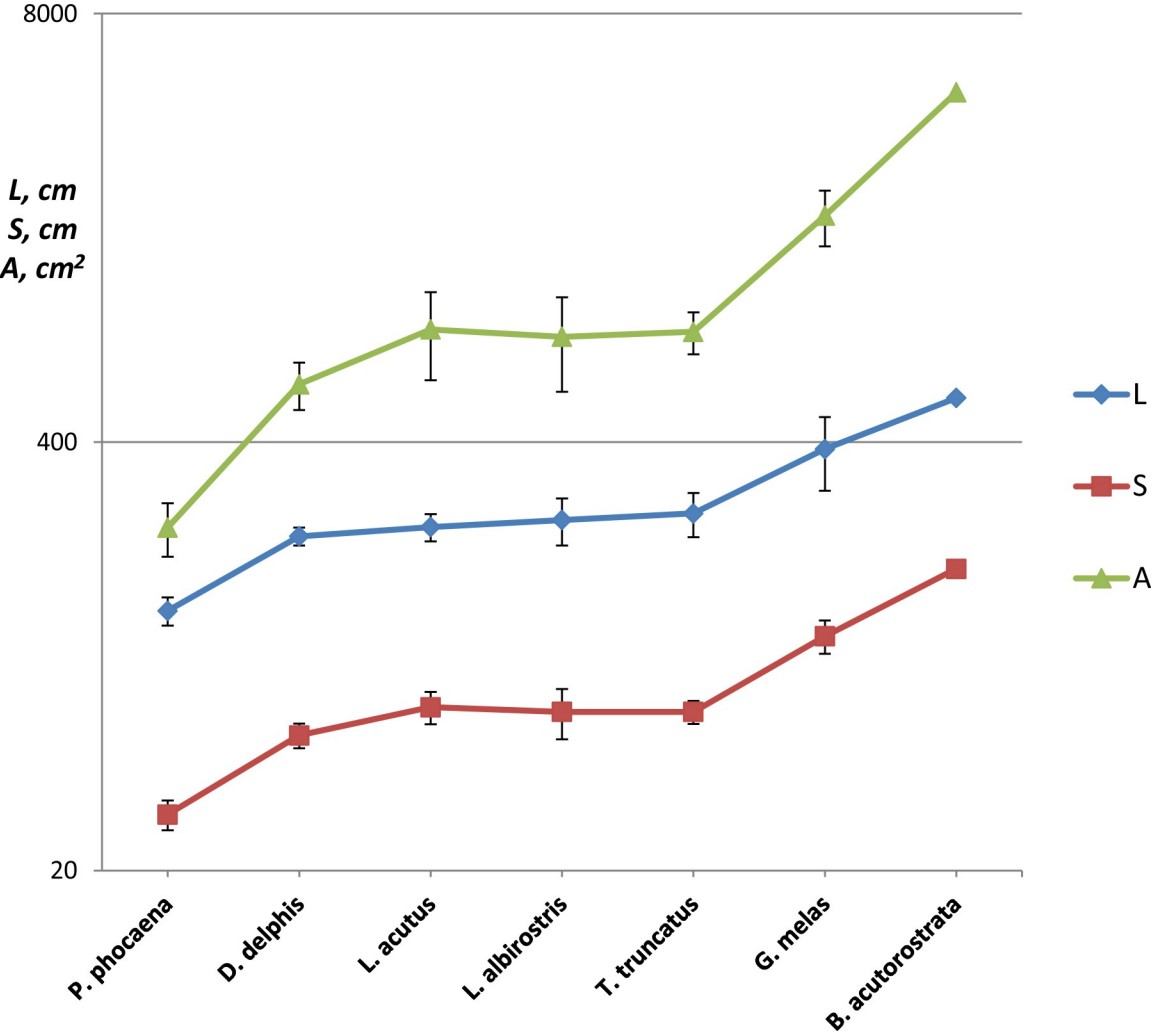

**Fig 8. Species-specific differences in the length of the body *L* cm, span of the fluke *S* cm and area of the fluke *A* cm², means ± SD.**

dimensionless parameters *AR*, *CI*, and Λ varied moderately (Table 2). Apart from these species, the dorsal fins of *P. phocoena* and *G. melas* had obvious distinctions both in dimensional and dimensionless parameters of the fin shape. A significant correlation was found between the body length and dimensional parameters of the dorsal fin planform (Table 3).

Cross-sections of dorsal fins showed similarity with the conventional symmetrical airfoils by Eppler, Selig, and NACA [46]. Cross-sections at the base of dorsal fins were comparable with the laminarized profiles E297 and E836 with the thickness ratio increased up to 15%, while the cross-sections at the fin tip had similarity with the E477, S1048, and NACA 0015 airfoils. A smooth transition in shape of the cross-section taken from the base and tip of the fin was observed. With a noticeable similarity to the geometry of the airfoils, all fin cross-sections had a distinctively thickened trailing edge (Fig 7).

Dimensional parameters of fin cross-sections *CL*, *MT* and *PMT* related with the fin size showed extreme values for *P. phocoena* and *B. acutorostrata* (S1–S3 Figs and S2 File). In contrast, the span-wise variation in these parameters appeared similar in all studied species. The geometry of the cross-sections at the base of the fin was more variable compared with the cross-sections located at the tip of the fin.

Apart from the dimensional parameters of the cross-sections, the dimensionless ones appeared to be more consistent (S4–S6 Figs). From the fin base to the fin tip, the *MT*, *%CL* decreased in *G. melas* and *L. albirostris*, varied slightly in *T. truncatus*, *L. acutus* and *B. acutorostrata*, and increased in *D. delphis* and *P. phocoena*. In the same direction, *PMT*, *%CL* varied slightly in P. phocoena, *L. albirostris* and *B. acutorostrata*, and increased in *D. delphis*, *L. acutus*, *T. truncatus* and *G. melas*.

Span-wise distribution of *LER*, *%CL* was revealed to be similar in all studied species. In general, this parameter increased from the fin base to two thirds of the fin span in all species, then varied slightly up to the fin tip. No species-specific distinctions were observed for this parameter, except for the *G. melas*, where average values of *LER*, *%CL* were significantly higher compared with other species.

## Shape and cross-sectional geometry of the flukes

With the variable size, *S* = 30 ± 3 (means ± SD) cm in *P. phocoena* and *S* = 165 cm in *B. acutorostrata*, the shape of the tail flukes appeared to be more uniform compared with the shape of the dorsal fins (Table 4). All species had a positive sweep of the leading and trailing edge where Λ of the leading edge varied from 27˚ in *B. acutorostrata* to 35 ± 1˚ (means ± SD) in *T. truncatus*. Distinctions were revealed between the *G. melas* and *B. acutorostrata* group having a trapezoidal shape for the tail flukes with high *AR*, and the Atlantic white sided dolphin, *T. truncatus* and *P. phocoena* group having swept back tips at the flukes and lower *AR* (Table 4). *D. delphis* and *L. albirostris* had a moderate sweep back at the tips of the flukes. Both *S* and *A* of the flukes showed a positive correlation with the length of the body (Fig 8). The length of the body and all

**Table 5. Correlation matrix of the dimensional and dimensionless parameters of the tail flukes in the Odontoceti species.**

| Variables | BL | S, cm | A, cm$^2$ | Λ, rad | AR |
|---|---|---|---|---|---|
| BL | 1 | 0.993 | 0.989 | -0.442 | 0.945 |
| S, cm | 0.993 | 1 | 0.995 | -0.419 | 0.961 |
| A, cm$^2$ | 0.989 | 0.995 | 1 | -0.439 | 0.961 |
| Λ, rad | -0.442 | -0.419 | -0.439 | 1 | -0.619 |
| AR | 0.945 | 0.961 | 0.961 | -0.619 | 1 |

Values in bold are different from 0 with a significance level alpha = 0.05.

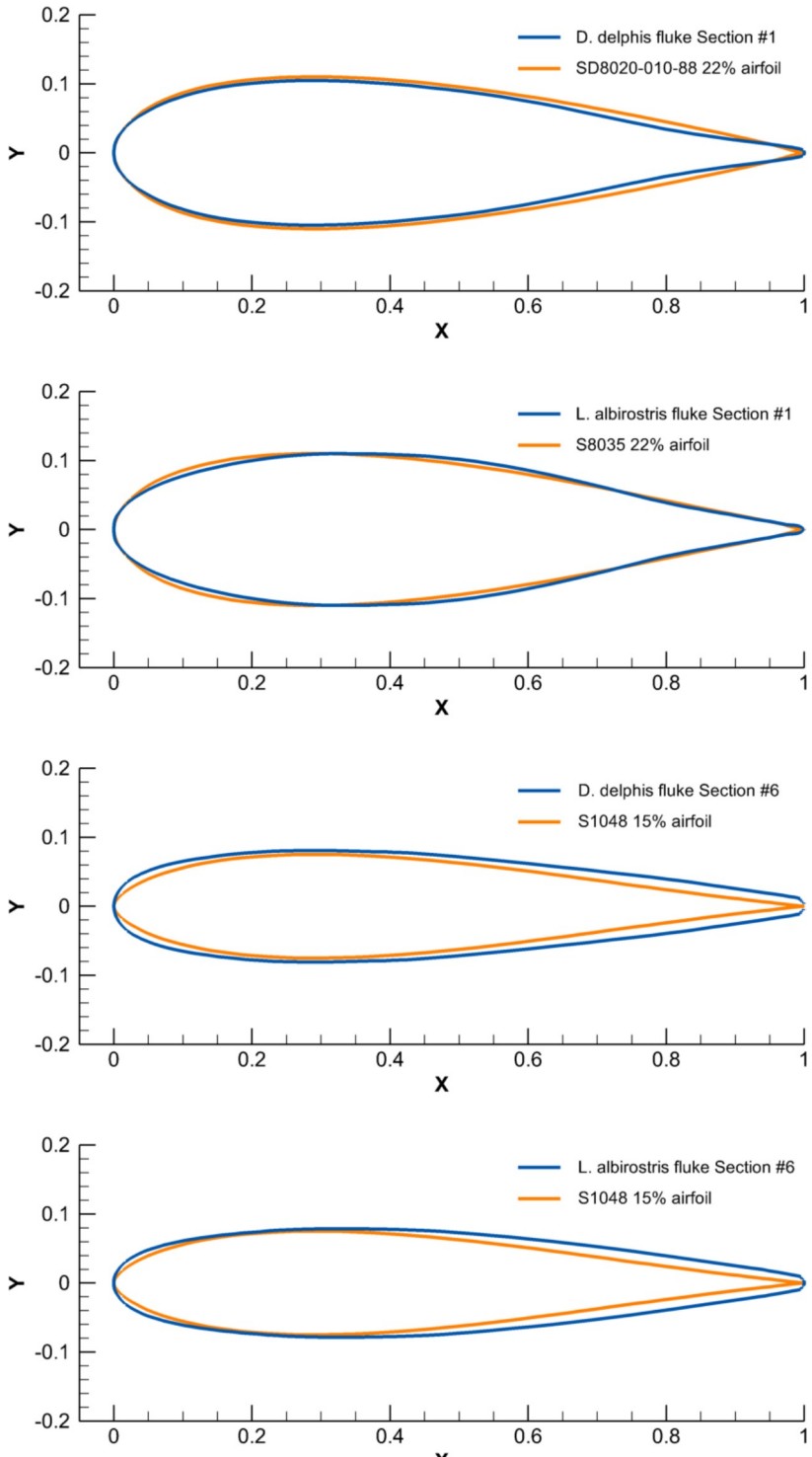

**Fig 9. Comparison of the chord-normalized profile coordinates of cross-sections taken at the base and top of tail flukes of the selected species with the conventional airfoils.**

parameters of the fin shape were found to be correlated except for the sweep Λ of the fin which showed a negative correlation with *AR* (Table 5).

**Table 6. ANOVA table for the dimensionless parameter MT%CL of the dorsal fin and tail fluke cross-sections with the independent factors fin type (Factor 1), section # (Factor 2), species (Factor 3) and body length (Factor 4).**

| Factors | Source | DF | Sum of squares | Mean squares | F | Pr > F | DF | R² | Adjusted R² |
|---|---|---|---|---|---|---|---|---|---|
| 1 | Model | 1 | 481 | 481 | 78 | <0.0001 | 448 | 0.148 | 0.146 |
| | Error | 448 | 2770 | 6 | | | | | |
| | Corrected | 449 | 3251 | | | | | | |
| 2 | Model | 5 | 418 | 84 | 13 | <0.0001 | 444 | 0.129 | 0.119 |
| | Error | 444 | 2833 | 6 | | | | | |
| | Corrected | 449 | 3251 | | | | | | |
| 3 | Model | 5 | 198 | 40 | 6 | <0.0001 | 444 | 0.061 | 0.050 |
| | Error | 444 | 3054 | 7 | | | | | |
| | Corrected | 449 | 3251 | | | | | | |
| 4 | Model | 1 | 6 | 6 | 1 | 0.374 | 448 | 0.002 | 0.000 |
| | Error | 448 | 3246 | 7 | | | | | |
| | Corrected | 449 | 3251 | | | | | | |
| 1 x 2 | Model | 6 | 899 | 150 | 28 | <0.0001 | 443 | 0.277 | 0.267 |
| | Error | 443 | 2352 | 5 | | | | | |
| | Corrected | 449 | 3251 | | | | | | |

The cross-sections of the flukes had a resemblance with the Eppler, Selig, and NACA conventional symmetrical airfoils [24]. Cross-sections at the base of dorsal fins were comparable with the SD8020-010-88 and S8035 airfoils, with the thickness ratio increasing to 22%, while the cross-sections at the fin tip had a similarity with the E477, S1048, and NACA 0015 airfoils with the thickness ratio increasing to 15% (Fig 9). All species had a similar span-wise distribution of *CL*, *MT* and *PMT*, this decreasing from the fin base to the fin tip (S7–S9 Figs and S2 File).

A similar pattern of *MT*, %*CL*, distribution in a span-wise direction was found in all studied species (S10–S12 Figs). In general, this parameter decreased from the base of the fluke to the fluke's tip. In *T. truncatus* and *B. acutorostrata* it slightly increased at the section located at the fluke's tip. The position of relative thickness *PMT*, %*CL* in the fluke's cross-sections varied

**Table 7. ANOVA table for the dimensionless parameter PMT%CL of the dorsal fin and tail fluke cross-sections with the independent factors fin type (Factor 1), section # (Factor 2), species (Factor 3) and body length (Factor 4).**

| Factors | Source | DF | Sum of squares | Mean squares | F | Pr > F | DF | R² | Adjusted R² |
|---|---|---|---|---|---|---|---|---|---|
| 1 | Model | 1 | 58736 | 58736 | 559 | <0.0001 | 448 | 0.555 | 0.554 |
| | Error | 448 | 47095 | 105 | | | | | |
| | Corrected | 449 | 105832 | | | | | | |
| 2 | Model | 5 | 397 | 79 | 0 | 0.892 | 444 | 0.004 | -0.007 |
| | Error | 444 | 105435 | 237 | | | | | |
| | Corrected | 449 | 105832 | | | | | | |
| 3 | Model | 5 | 24769 | 4954 | 27 | <0.0001 | 444 | 0.234 | 0.225 |
| | Error | 444 | 81063 | 183 | | | | | |
| | Corrected | 449 | 105832 | | | | | | |
| 4 | Model | 1 | 7150 | 7150 | 32 | <0.0001 | 448 | 0.068 | 0.065 |
| | Error | 448 | 98682 | 220 | | | | | |
| | Corrected | 449 | 105832 | | | | | | |
| 1 x 2 | Model | 6 | 59133 | 9856 | 93 | <0.0001 | 443 | 0.559 | 0.553 |
| | Error | 443 | 46698 | 105 | | | | | |
| | Corrected | 449 | 105832 | | | | | | |

within the range of 23–36%CL with maximal and minimal values in *G. melas* and *P. phocoena* respectively. Average values of *LER*, *%CL* decreased from the base of the fluke to the mid-span and then gradually increased to the tip of the fluke. This trend was different in B. acutorostrata where the *LER*, *%CL* decreased at the section located at the tip of the flukes.

## Analysis of the cross-sectional geometry of the dorsal fin and flukes

To verify our hypotheses on a generic wing design for the fins, we used ANOVA to examine how the variation in the dimensionless parameters of a fin cross-section *MT*, *%CL*, *PMT*, *% CL*, and *LER*, *%CL* was related with the fin type, position on the fin, species and body length (Tables 6–8).

The variation in all dimensionless parameters of the fin cross-sections was found to be related primarily with the fin type (Factor 1), this explaining most of the variability. The position on the fin (Factor 2) had an significant effect on the span-wise distribution of the *MT*, *% CL*, and *LER*, *%CL* dimensionless parameters. The interaction effect of the fin type (Factor 1) and position on the fin (Factor 2) was also present and had a maximal value compared with other possible combinations of factors. Species (Factor 3) and body length (Factor 4) had a smaller or zero effect on the dimensionless parameters associated with thickness, i.e., *MT*, *% CL* and *PMT*, *%CL* and zero effect on *LER*, *%CL*.

The pattern of cross-sectional geometry variation and the relationship between the dimensionless parameters of fins were examined with the PCA (Fig 10A–10C). The first two components explained 84.5% of the variability. The first component that can be interpreted as the shape of the foil, explained 55.45% of the variability and described a variation in the *LER%CL* and *PMT%CL* (Fig 10A). This variation presented two distinctive patterns of cross-sectional geometry: The tail fluke's sections with the higher *LER%CL* and shifted forward *PMT%CL*, and the dorsal fin sections with lower *LER%CL* and shifted backward *PMT%CL* (Fig 11B). It was found that the cross-sections located at the base of the dorsal fin and tail flukes had the most distinctive hydrofoil design with extreme values for *LER%CL* and *PMT%CL* (Figs 10C and 11). The second component that can be interpreted as the thickness of the foil, explained 29.06% of the variability and described a variation from thick to thin cross-sections. *LER%CL*

**Table 8. ANOVA table for the dimensionless parameter LER%CL of the dorsal fin and tail fluke cross-sections with the independent factors fin type (Factor 1), section # (Factor 2), species (Factor 3) and body length (Factor 4).**

| Factors | Source | DF | Sum of squares | Mean squares | F | Pr > F | DF | R² | Adjusted R² |
|---|---|---|---|---|---|---|---|---|---|
| 1 | Model | 1 | 367 | 367 | 789 | **<0.0001** | 448 | 0.638 | 0.637 |
| | Error | 448 | 208 | 0 | | | | | |
| | Corrected | 449 | 575 | | | | | | |
| 2 | Model | 5 | 44 | 9 | 7 | **<0.0001** | 444 | 0.076 | 0.066 |
| | Error | 444 | 531 | 1 | | | | | |
| | Corrected | 449 | 575 | | | | | | |
| 3 | Model | 5 | 13 | 3 | 2 | 0.063 | 444 | 0.023 | 0.012 |
| | Error | 444 | 562 | 1 | | | | | |
| | Corrected | 449 | 575 | | | | | | |
| 4 | Model | 1 | 0 | 0 | 0 | 0.931 | 448 | 0.000 | -0.002 |
| | Error | 448 | 575 | 1 | | | | | |
| | Corrected | 449 | 575 | | | | | | |
| 1 x 2 | Model | 6 | 410 | 68 | 184 | **<0.0001** | 443 | 0.714 | 0.710 |
| | Error | 443 | 165 | 0 | | | | | |
| | Corrected | 449 | 575 | | | | | | |

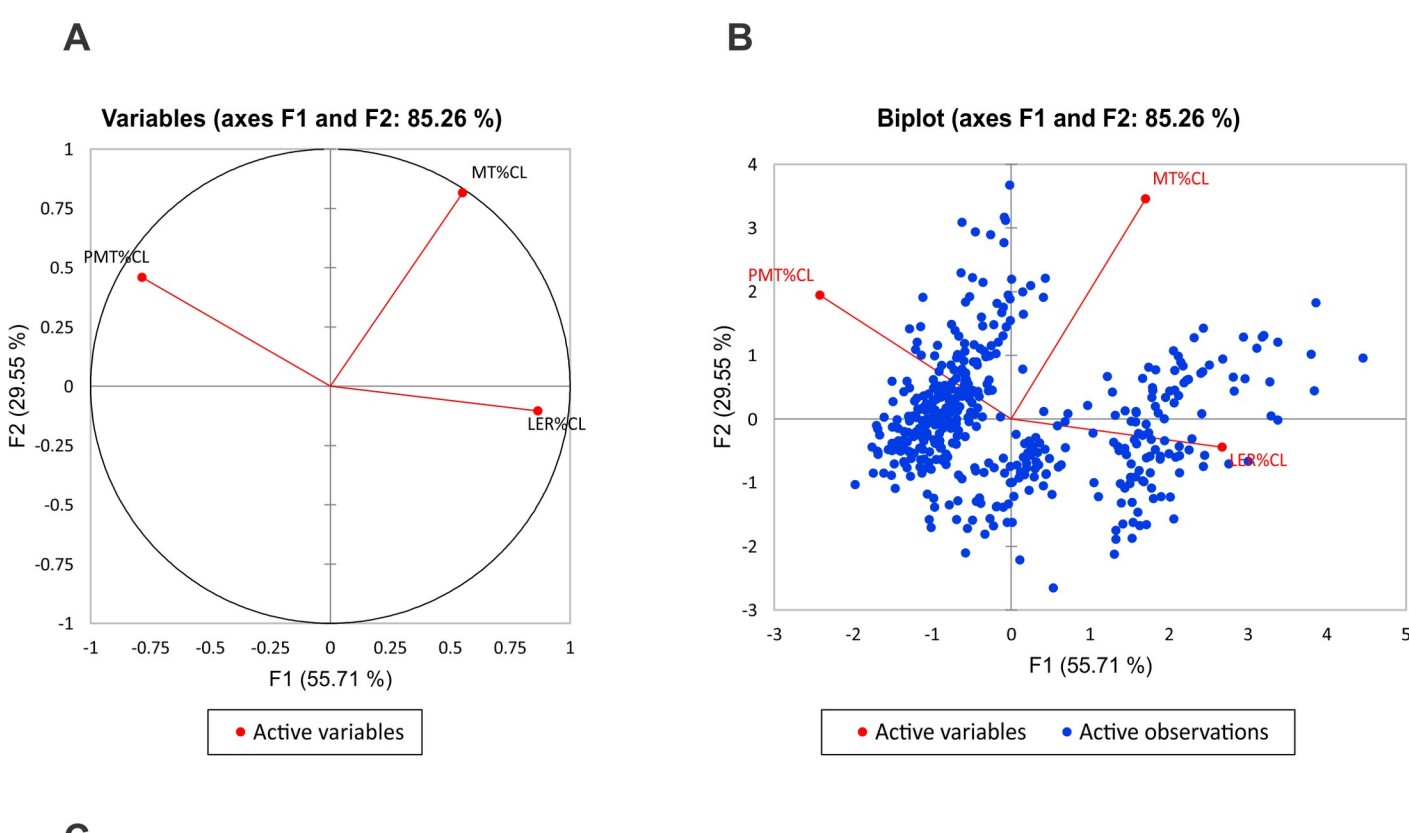

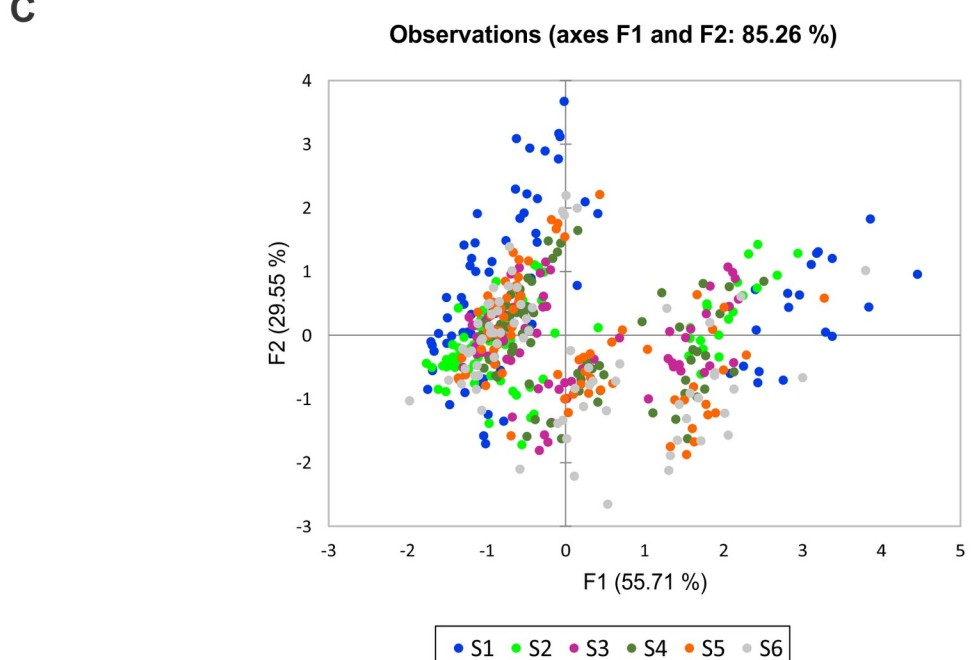

**Fig 10. Principal Component Analysis of the dimensionless parameters of the dorsal fin and tail fluke cross-sections of all species.** A–A negative relationship between *LER%CL* and *PMT%CL* is shown, while *MT%CL* is unrelated with the *LER%CL* and has a slight positive correlation with the *PMT%CL*. B–Separation between the dorsal fin and fluke's cross-sections based on the difference in *LER%CL* and *PMT%CL*. C–Cross-sections located at the base of the fins (blue dots) indicate distinctive hydrofoil design with extreme values for *LER%CL* and *PMT%CL*.

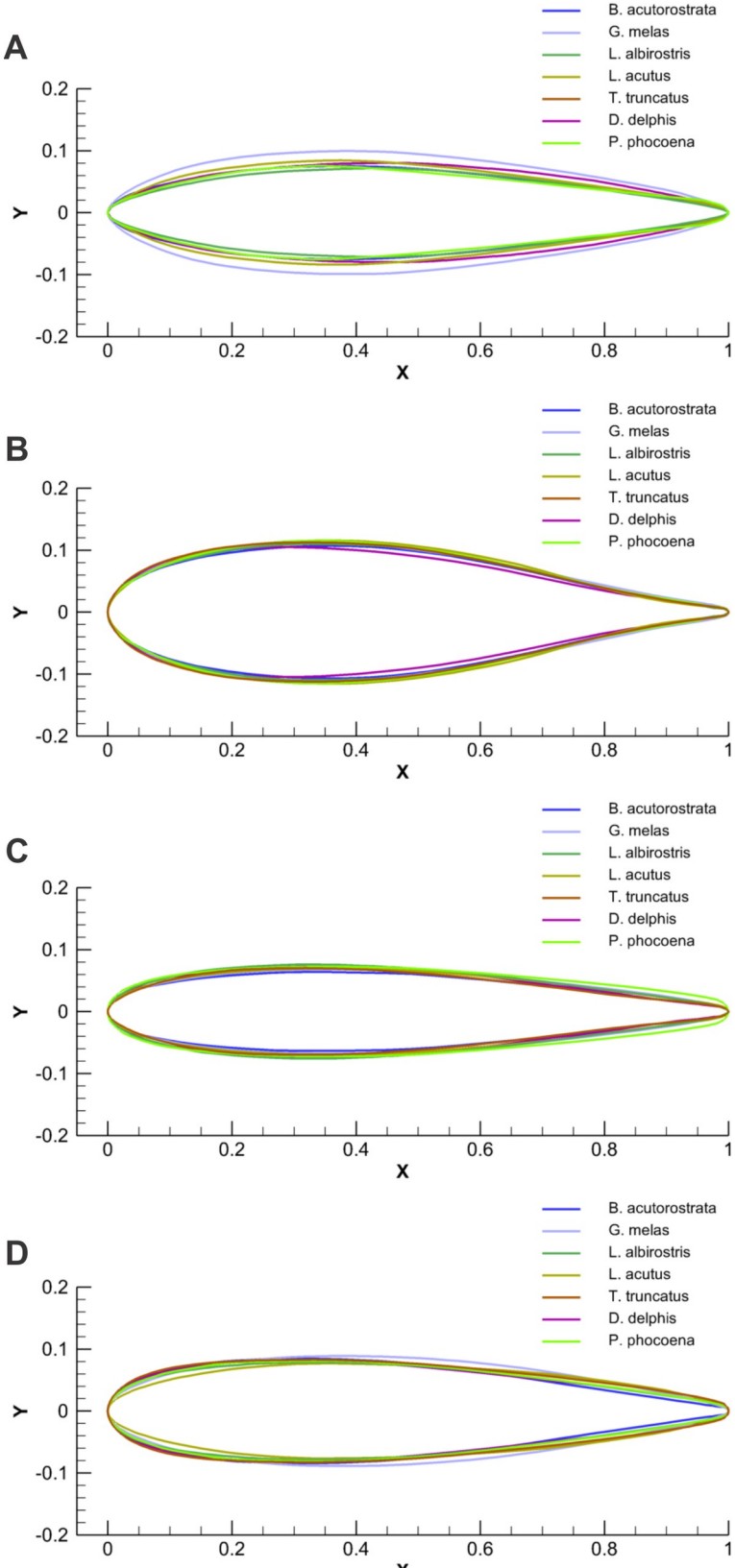

**Fig 11. Comparison of the chord-normalized profile coordinates of cross-sections taken at the base and top of the dorsal fin and tail flukes.** A—Cross-sections taken at the base of the dorsal fin. B—Cross-sections taken at the base of the tail fluke. C—Cross-sections taken at the top of the dorsal fin. D—Cross-sections taken at the top of the tail fluke.

had a negative correlation with the *PMT%CL* and a slight positive correlation with *MT%CL*, while no correlation was revealed between *PMT%CL and MT%CL*.

## Hydrodynamic characteristics of the cross-sections of dorsal fin

At slow speed of swimming 2 m/sec, the drag coefficient *Cd* of the fin cross-sections increased from the fin base to the fin tip (S13 Fig). At fast speed of swimming 8 m/sec, the *Cd* slightly decreased, while a span-wise pattern of *Cd* distribution remained the same (S14 Fig).

Cross-sections located at the base of the fin as well as the 297 and 836 airfoils had comparable average values of *Cd* calculated for $\alpha = 0°$ (Table 9). Pressure distribution had a sharp negative pressure gradient at the leading edge with minimal *Cp* values at 15–20% of the chord length. Both in the cross-sections and airfoils, the region from 20 to 60% of the chord length was characterized by zero or a slightly positive pressure gradient. The average values of *Cd* of cross-sections located at the tip of the fin were comparable with the *Cd* of 477 and S1048

**Table 9. Comparison of the hydrodynamic characteristics of the dorsal fin cross-sections with the appropriate airfoils.**

| Species | Section # | Re number | Cd 2 /sec | Re number | Cd 8 /sec | min Cp |
|---|---|---|---|---|---|---|
| *P. phocoena* | 1 | 3.60E+05 | 0.01 | 1.50E+06 | 0.0073 | -0.383 |
| *D. delphis* | 1 | 4.40E+05 | 0.0083 | 1.80E+06 | 0.0065 | -0.398 |
| *L. acutus* | 1 | 5.30E+05 | 0.0089 | 2.10E+06 | 0.007 | -0.414 |
| *L. albirostris* | 1 | 5.50E+05 | 0.0101 | 2.90E+06 | 0.0073 | -0.398 |
| *T. truncatus* | 1 | 5.30E+05 | 0.0101 | 2.70E+06 | 0.0077 | -0.456 |
| *G. melas* | 1 | 9.20E+05 | 0.011 | 3.70E+06 | 0.0081 | -0.514 |
| *B. acutorostrata* | 1 | 6.99E+05 | 0.0088 | 2.80E+06 | 0.0066 | -0.438 |
| M | | **5.76E+05** | **0.0096** | **2.50E+06** | **0.0072** | **-0.429** |
| SD | | 1.84E+05 | 0.000952 | 7.50E+05 | 0.0006 | 0.045 |
| **Airfoils** | | | | | | |
| E297* | | 5.76E+05 | 0.0076 | 2.50E+06 | 0.0062 | -0.417 |
| E836* | | 5.76E+05 | 0.0067 | 2.50E+06 | 0.0052 | -0.403 |
| **Species** | | | | | | |
| *P. phocoena* | 6 | 1.40E+05 | 0.0135 | 5.70E+05 | 0.009 | -0.515 |
| *D. delphis* | 6 | 1.70E+05 | 0.0136 | 6.60E+05 | 0.0103 | -0.572 |
| *L. acutus* | 6 | 1.80E+05 | 0.0132 | 7.10E+05 | 0.0094 | -0.507 |
| *L. albirostris* | 6 | 1.80E+05 | 0.0111 | 7.10E+05 | 0.0092 | -0.524 |
| *T. truncatus* | 6 | 2.10E+05 | 0.0124 | 7.70E+05 | 0.0092 | -0.488 |
| *G. melas* | 6 | 3.20E+05 | 0.0106 | 1.30E+06 | 0.0085 | -0.443 |
| *B. acutorostrata* | 6 | 2.15E+05 | 0.0117 | 8.60E+05 | 0.0083 | -0.399 |
| M | | **2.02E+05** | **0.0123** | **7.97E+05** | **0.0091** | **-0.493** |
| SD | | 5.77E+04 | 0.0012 | 2.39E+05 | 0.0007 | 0.057 |
| **Airfoils** | | | | | | |
| E477* | | 2.00E+05 | 0.0108 | 7.97E+05 | 0.079 | -0.483 |
| S1048* | | 2.00E+05 | 0.0112 | 7.97E+05 | 0.0082 | -0.531 |

*Modified profile with thickness ratio increased to 15%.

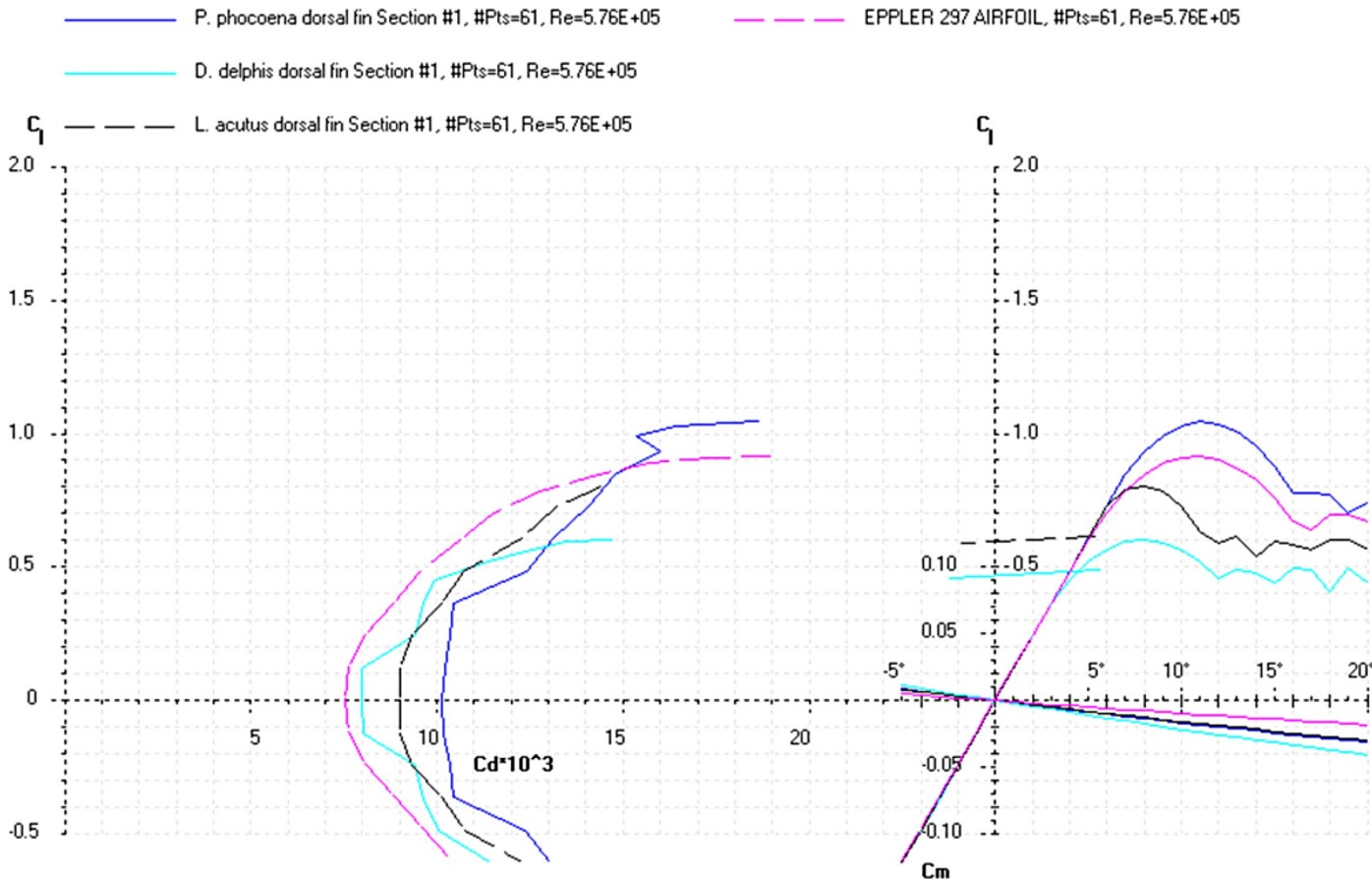

**Fig 12. Drag polar diagram of lift CL vs drag Cd.** Calculated for the cross-sections taken at the base of the dorsal fin of the *P. phocoena*, *D. delphis*, *L. acutus* and Eppler 297 airfoil at the averaged *Re* 5.76E+05 for the fin cross-sections (Table 9).

airfoils calculated under the same conditions (Table 9). Regarding peculiarity of pressure distribution, there was an absence of zero or a slightly positive pressure gradient. A change in the sign of gradient occurred at 20–25% of the chord length.

The *Cd*, *Cl* and *Cm* coefficients of the cross-sections at the base and top of the fin (S3 File) and the appropriate airfoils were calculated for the range of α from 0 to 20˚ at the averaged *Re* numbers (Table 9) and plotted in the drag polar diagram (Figs 12–14 and S4 and S5 Files). Cross-sections of the dorsal fin located at the base of the fin possessed the maximum *L/D* ratio for the range of α = 4˚ for the *D. delphis* to 9˚ for the *P. phocoena* and the least stall angle 8˚ (Figs 12–14). These cross -sections had more abrupt stall characteristics compared with the mid-span and fin top location. Cross-sections of the dorsal fin located at the top of the fin possessed the maximum *L/D* ratio for the range of α = 8˚ for the *P. phocoena* and *G. melas* to 13˚ for the *T. truncatus* (S4 and S5 Files).

## Hydrodynamic characteristics of the cross-sections of tail flukes

At slow speed of swimming 2 m/sec, the *Cd* of the fluke cross-sections decreased from the fluke's base to 17–34% of the fluke's span and then gradually increased to the fluke tip (S15 Fig). At fast speed of swimming 8 m/sec, the *Cd* of the fluke's cross-sections decreased, while a span-wise pattern of *Cd* distribution remained the same (S16 Fig).

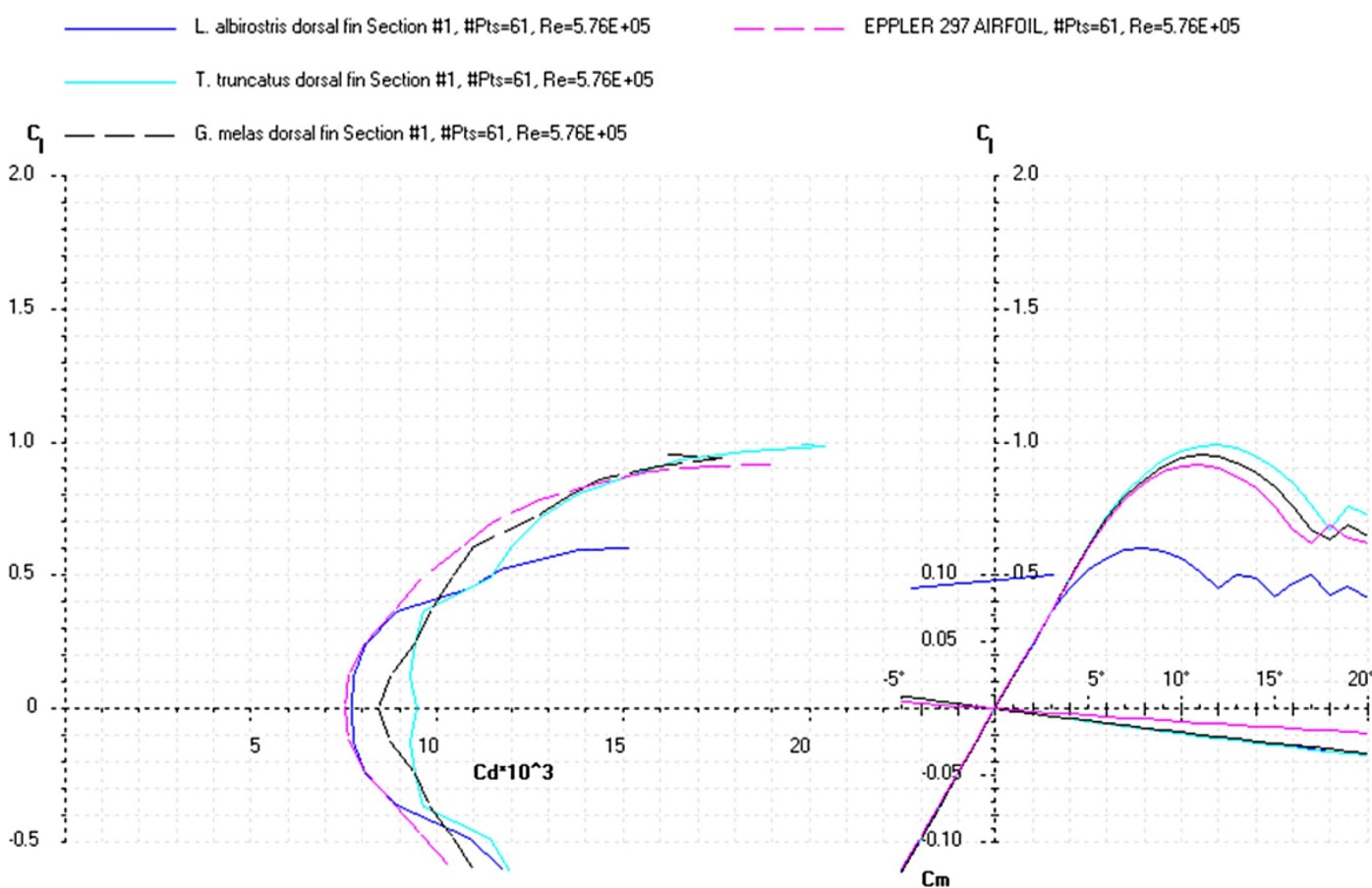

**Fig 13. Drag polar diagram of lift CL vs drag Cd.** Calculated for the cross-sections taken at the base of the dorsal fin of the *L. albirostris*, *T. truncatus*, *G. melas* and Eppler 297 airfoil at the averaged *Re* 5.76E+05 for the fin cross-sections (Table 9).

Cross-sections located at the base of the fin as well as the SD8020 and S8035 airfoils had comparable average values of *Cd* calculated for α = 0˚ (Table 10). The peculiarity of these profiles was the absence of zero or a slightly positive pressure gradient. Due to higher curvature of cross-sections, the length of transition zone varied within a narrow range from 5 to 8% of *CL*. The specific shape of cross-sections at the base of the fluke had the lowest minimal values of pressure coefficient *Cp* -0.746 that exceeded the value of -0.514 of the same parameter of cross-sections located at the base of the dorsal fin. The shape of cross-sections located at the top of the flukes was quite similar to the cross-sections of the dorsal fin made at the same locations (Tables 9 and 10). As a consequence, the average *Cd* as well as pressure distribution of the cross-sections at that location differed slightly.

The *Cd*, *Cl* and *Cm* coefficients of the cross-sections taken at the base and top of the fin (S6 File) and the appropriate airfoils were calculated for the range of α from 0 to 20˚ at the averaged *Re* numbers (Table 10) and plotted in the drag polar diagram (Figs 15–17 and S5 and S6 Files). Cross-sections of the dorsal fin located at the base of the fin possessed the maximum *L/D* ratio for the range of α = 8˚ for the *P. phocoena* and *T. truncatus* to 9˚ for the other species and the least stall angle 15˚ (Figs 15–17). These cross-sections had relatively gradual stall characteristics. Cross-sections of the dorsal fin located at the top of the fin possessed the maximum *L/D* ratio for the range of α = 8˚ for the *G. melas* to 13˚ for the *P. phocoena* (S7 and S8 Files).

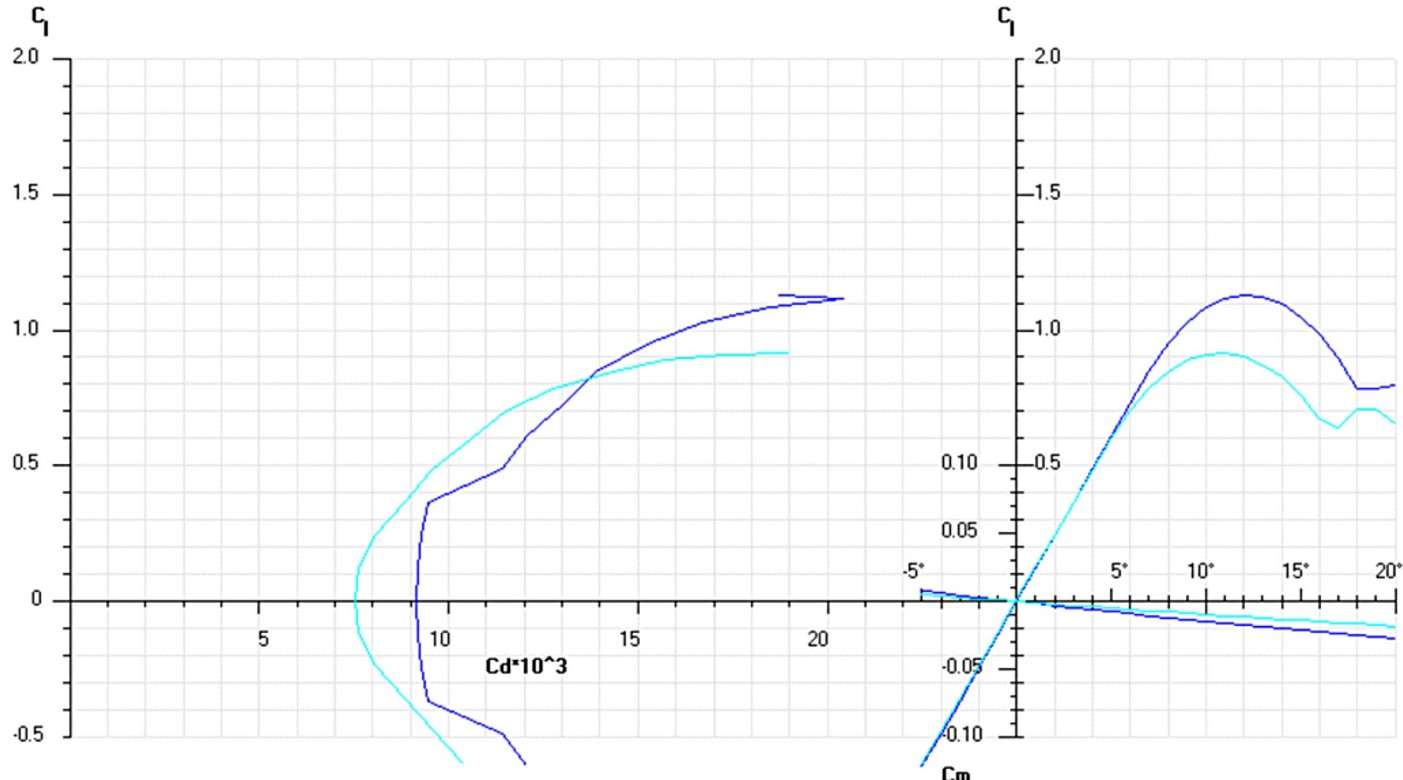

**Fig 14. Drag polar diagram of lift CL vs drag Cd.** Calculated for the cross-sections taken at the base of the dorsal fin of the *B. acutorostrata* and Eppler 297 airfoil at the averaged *Re* 5.76E+05 for the fin cross-sections (Table 9).

## Comparison of the shape and cross-sections of the dorsal fin and tail flukes

The shape of the dorsal fin and tail flukes had distinctive features both on the planform and cross-sectional levels. In general, the dorsal fins had more swept-back planform with low *AR* compared with the tail flukes (Fig 18). The Λ vs *AR* scatterplot showed different trends in variation of the fin planform. The planform of dorsal fin had noticeable variation along the Λ axis with the extreme values in *G. melas* and *P. phocoena*. The tail flukes had moderate variation along the both axes except of the high *AR* flukes of the *G. melas* and *B. acutorostrata* (Fig 18).

At a similar cross-sectional design in both fins, two distinctive patterns of cross-sectional geometry were revealed in the dorsal fin and tail fluke. In general, the tail fluke's sections were thicker, with the higher *LER%CL* and shifted forward *PMT%CL*, while the dorsal fin sections were thinner, with lower *LER%CL* and shifted backward *PMT%CL*. The main difference as in geometry as in hydrodynamic characteristics of the cross-sections was found regarding the fin-body junction. Cross-sections of the dorsal fin located at this region possessed the maximum *L/D* ratio for the range of α = 4–9˚ and the least stall angle 8˚ (Figs 12–17). These cross-sections had more abrupt stall characteristics compared with the mid-span and fin tip location. Cross-sections of the tail fluke taken at the same location had the maximum *L/D* ratio for the range of α = 8–9˚, the least stall

**Table 10. Comparison of the hydrodynamic characteristics of the tail fluke cross-sections with the appropriate airfoils.**

| Species | Section # | Re number | Cd 2 /sec | Re number | Cd 8 /sec | min Cp |
|---|---|---|---|---|---|---|
| *P. phocoena* | 1 | 2.11E+05 | 0.0132 | 8.45E+05 | 0.0095 | -0.746 |
| *D. delphis* | 1 | 3.19E+05 | 0.0127 | 1.28E+06 | 0.0094 | -0.753 |
| *L. acutus* | 1 | 4.23E+05 | 0.011 | 1.69E+06 | 0.0076 | -0.699 |
| *L. albirostris* | 1 | 4.68E+05 | 0.0114 | 1.87E+06 | 0.0081 | -0.697 |
| *T. truncatus* | 1 | 4.60E+05 | 0.012 | 1.84E+06 | 0.0091 | -0.742 |
| *G. melas* | 1 | 5.04E+05 | 0.0104 | 2.02E+06 | 0.0078 | -0.695 |
| *B. acutorostrata* | 1 | 9.57E+05 | 0.0085 | 3.83E+06 | 0.006 | -0.636 |
| M | | **4.77E+05** | **0.0113** | **1.91E+06** | **0.0082** | **-0.71** |
| SD | | 2.34E+05 | | 9.38E+05 | | |
| **Airfoils** | | | | | | |
| S8035* | | 4.77E+05 | 0.0098 | 1.91E+06 | 0.0074 | -0.8 |
| SD8020* | | 4.77E+05 | 0.0098 | 1.91E+06 | 0.0075 | -0.818 |
| **Species** | | | | | | |
| *P. phocoena* | 6 | 1.06E+05 | 0.0154 | 4.23E+05 | 0.0111 | -0.49 |
| *D. delphis* | 6 | 1.18E+05 | 0.0149 | 4.74E+05 | 0.0116 | -0.571 |
| *L. acutus* | 6 | 1.48E+05 | 0.0136 | 5.94E+05 | 0.0094 | -0.433 |
| *L. albirostris* | 6 | 1.64E+05 | 0.0142 | 6.57E+05 | 0.0108 | -0.501 |
| *T. truncatus* | 6 | 1.77E+05 | 0.0159 | 7.09E+05 | 0.0117 | -0.59 |
| *G. melas* | 6 | 1.50E+05 | 0.0122 | 6.02E+05 | 0.0089 | -0.489 |
| *B. acutorostrata* | 6 | 2.74E+05 | 0.0121 | 1.09E+06 | 0.0097 | -0.591 |
| M | | **1.63E+05** | **0.014** | **6.50E+05** | **0.0105** | **-0.524** |
| SD | | 5.49E+04 | | 2.20E+05 | | |
| **Airfoils** | | | | | | |
| E477** | | 1.63E+05 | 0.0113 | 6.50E+05 | 0.0084 | -0.484 |
| S1048** | | 1.63E+05 | 0.0121 | 6.50E+05 | 0.0088 | -0.533 |

*Modified profile with thickness ratio increased to 22%.

**Modified profile with thickness ratio increased to 15%.

angle 15˚ (Figs 12–17), and more gradual stall characteristics. Span-wise lift distribution $C^*Cl_{max}$ calculated with the illustrative purpose for the *D. delphis* presented similar pattern on the cross-sections of both appendages with decreasing $L$ from the base of the fin to the fin top (Fig 20).

Cross-sections of both appendages exhibited the lowest drag over a narrow range of angle of attack called the "drag bucket" (Figs 12–17). The term "drag bucket" is used to describe the shape of a drag curve showing *Cd* against α, where the drag curve shows an extended flat bottom of the curve, i.e., bucket-shaped [17]. The shape and position of the drag bucket were solely determined by the cross-sectional geometry. Generally, the width of the drag bucket decreased from the base to mid-span of the fin and then increased to the fin top, according to the span-wise distribution of *LER*, *%CL*, *MT*, *%CL* and the *PMT*, *%CL*.

## Discussion

### Morphology and hydrodynamics of fins

In this paper, we tested our hypotheses about a generic wing design of the fins of cetaceans acting as active and passive control surfaces in stabilizing the straight-line swimming, turning control and thrust production [10,18,21]. Generic shape of the fins presents a wing-like

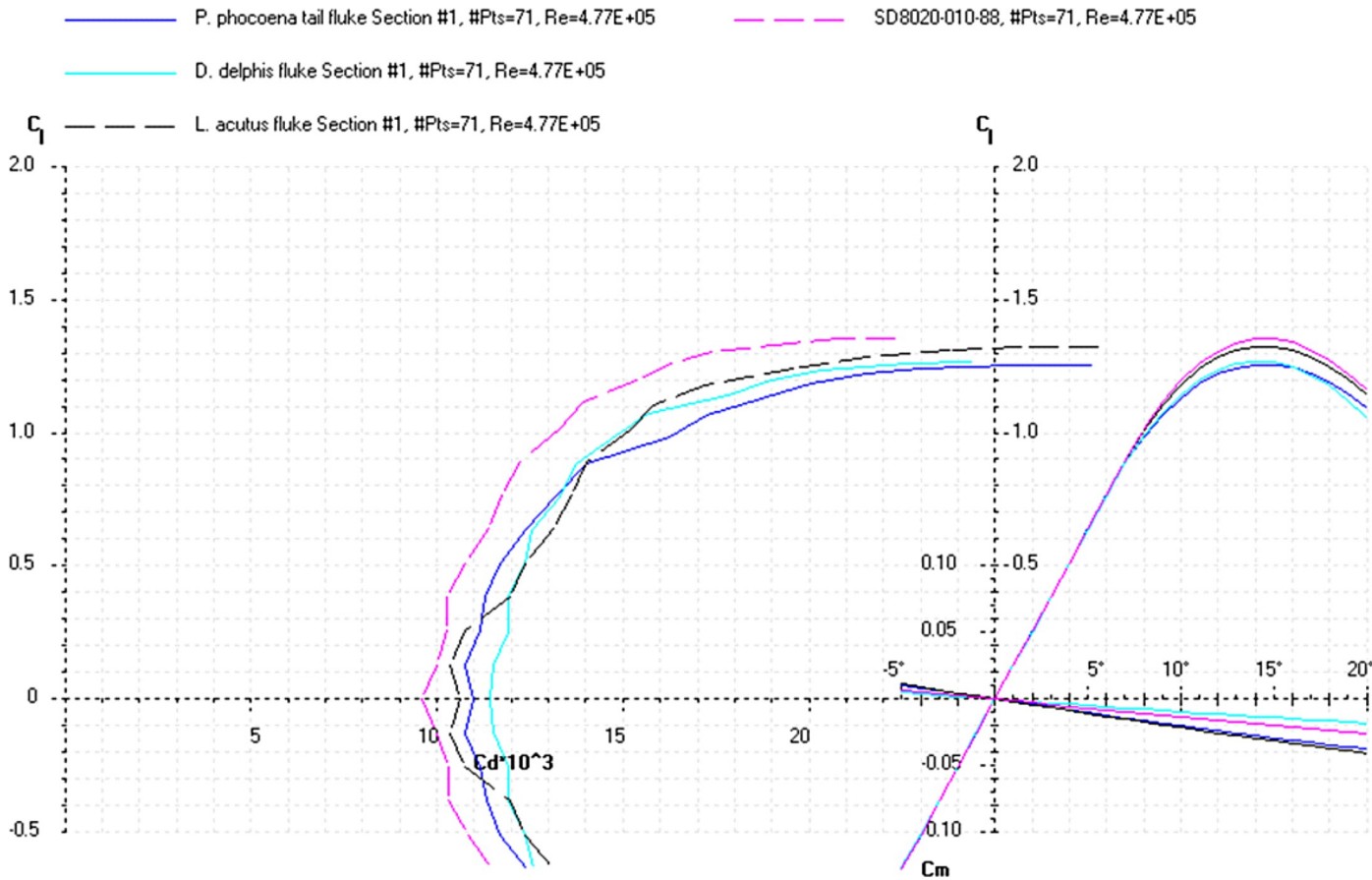

**Fig 15. Drag polar diagram of lift CL vs drag Cd.** Calculated for the cross-sections taken at the base of the tail flukes of the *P. phocoena*, *D. delphis*, *L. acutus* and SD8020 airfoil at the averaged *Re* 4.77E+05 for the fluke's cross-sections (Table 10).

planform and the foil-like cross-sectional geometry [3,11,13–16] which was hypothesized to be invariant with respect to the body length and species. Both the planform and cross-sectional level of a generic fin shape had specific trends in variation (Figs 18 and 19) and different scaling with the length of the body (Figs 6 and 8 and Tables 3 and 5–8). In all species, the planform dimensions of fins were related with the length of the body in agreement with the previously published data [3,7,8] that is associated with the amount of generated thrust, lift, and drag for locomotion, stability and maneuverability control [21]. The fin planform had the specific patterns (Fig 18) attributed to the primary function as a fixed or flapping hydrofoil [18,21]. At a cross-sectional level, the dimensionless parameters of the fin cross-sections were found to be largely invariant with respect to the body length and species (Tables 6–8), thus supporting our **Hypothesis I**. It was also found that variation in dimensionless parameters of the fin cross-sections (Figs 10 and 19) is strongly associated with their primary function as a fixed or flapping hydrofoil (Tables 6–8), thus supporting our **Hypothesis II**.

The shape of both fins had an aerodynamic twist, i.e., a gradual modification of the cross-sectional geometry in a spanwise direction [17,47]. In combination with the generic swept-back tapered planform, this is associated with lift distribution so that more lift can be generated at the wing root and less towards the wingtip (Fig 20). This pattern in span-wise lift distribution causes a reduction in the strength of the wing tip vortices and a reduction in lift-induced drag [48]. Both fins are also characterized by a smooth filleted joint with the dolphin's

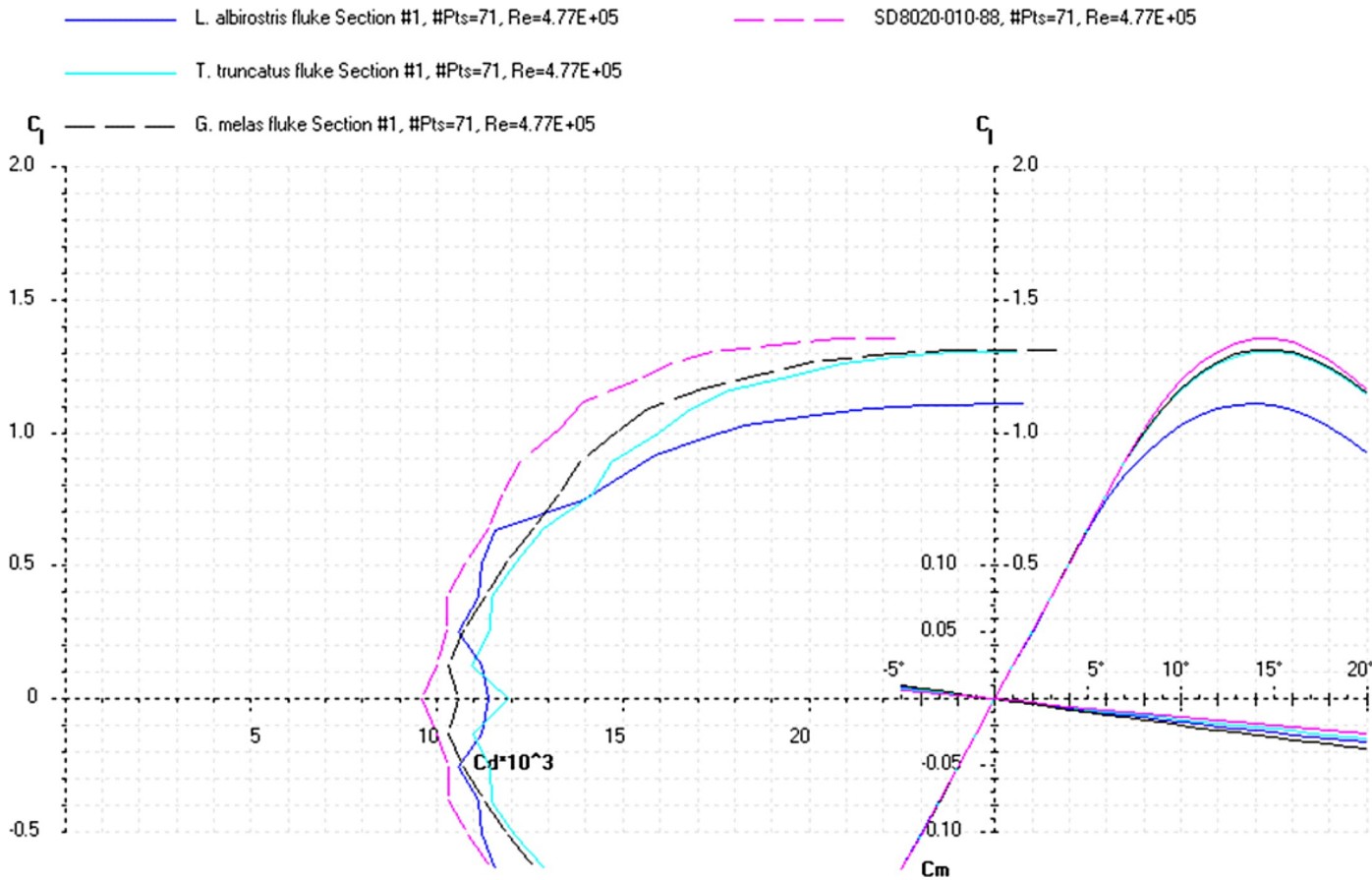

**Fig 16. Drag polar diagram of lift CL vs drag Cd.** Calculated for the cross-sections taken at the base of the tail flukes of the *L. albirostris*, *T. truncatus*, *G. melas* and SD8020 airfoil at the averaged *Re* 4.77E+05 for the fluke's cross-sections (Table 10).

body. This external morphology feature is associated with decreasing the interference drag, appearing when surfaces at angles to one another simulate turbulence in the region of the joint as can be observed in the intersection of the fuselage and wing in aviation [49].

The aerodynamic twist pattern was found to be different in the dorsal fin and tail flukes. To gain insight into the physical point of variation in cross-sectional geometry, we compared it with the engineered foils performing a similar function. While the top cross-sections were similar in both fins and comparable with the 477 and S1048 airfoils (Figs 7, 9 and 12–17), the root cross-sections were different. The root cross-sections of the dorsal fin showed a similarity both in shape and calculated *Cd*, *Cl*, *Cm*, and *Cp* coefficients with the modified versions of the E297 and E836 airfoils used for the yacht keels and rudders, and optimized for the maximum *L/D* ratio within a range of $\alpha$ = 5–7° (Figs 7, 9 and 12–17). The distinctive feature of the dorsal fin was its extended root section. This design is widespread in aircraft vertical stabilizers, e.g., in Boeing 737, as well as in yacht keels, and is associated with least interference drag, highest performance within narrow range of $\alpha$, but sudden stall characteristics outside the narrow lift-coefficient range [44,50].

The root cross-sections of the tail flukes had similarity both in shape and *Cd*, *Cl*, *Cm*, and *Cp* coefficients with the S8035 and SD8020 foils used for the aerobatic wings, having a more gradual stall pattern and allowing a greater angle of attack variations without producing excessive minimum pressure peaks at the leading edge, that can result in flow separation and cavitation

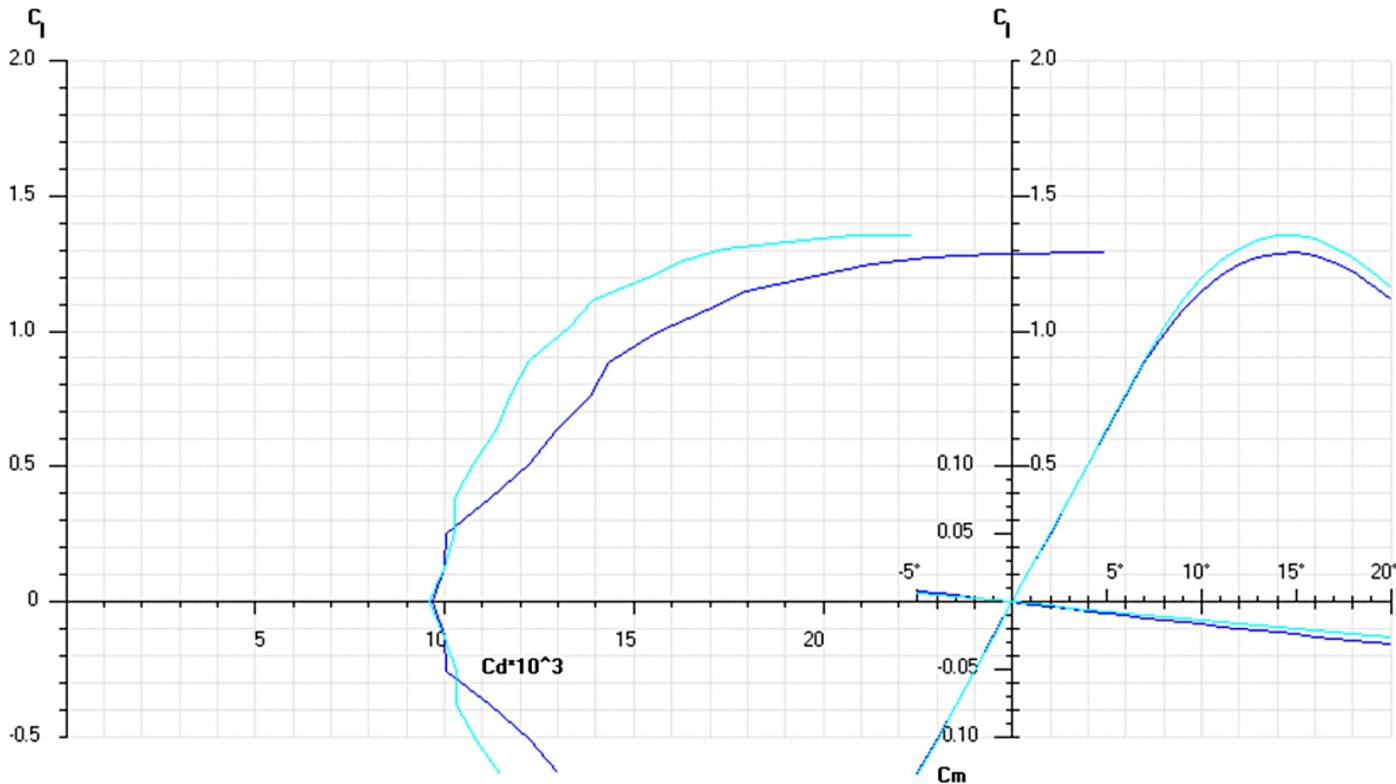

**Fig 17. Drag polar diagram of lift CL vs drag Cd.** Calculated for the cross-sections taken at the base of the tail flukes of the *B. acutorostrata* and SD8020 airfoil at the averaged *Re* 4.77E+05 for the fluke's cross-sections (Table 10).

[18]. The modified S8035 and SD8020 airfoils with thickness ratio increased to 22% are optimized for the maximum *L/D* ratio at α = 10–11˚, that falls within the range of observed α in dolphin flukes with mean values varying from 4.6 to 17.5˚ for *T. truncatus* [51], and maximum values varying from 22–24˚ in *Lagenorhynchus obliquidens* [29] to 30˚ in *T. truncatus* [51].

The distinctions revealed in the aerodynamic twist pattern of both fins indicate the fundamental difference between fixed and flapping hydrofoils. The fixed wing performance is limited by the angles of attack, causing stall condition, i.e., a dramatic decrease in the lift and increase drag. Flapping wings, on the contrary, can operate within a wider range of the angles of attack without any loss of efficiency because of dynamic stall caused by unsteady effects [22,24,52].

The wing-like shape of the dolphin fins presents an opportunity to compare it with the engineered lifting structures using standard wing and airfoil parameters. This comparison, though, should be made with caution. Unlike the engineered control surfaces, the biological structures are normally multifunctional rather than being optimized for a single function. The fins in cetaceans are involved in propulsion, stabilization, trim and maneuvering as well as in thermoregulation, and manifestation of sexual dimorphism in phenotypes in some species [21,53,54]. This assumes the fin shape as a trade-off between these different demands. In addition, the fins of cetaceans have specific constraints of structural strength and stiffness of the biological tissues that results in lower *AR* compared with wings or keels. Finally, the fins

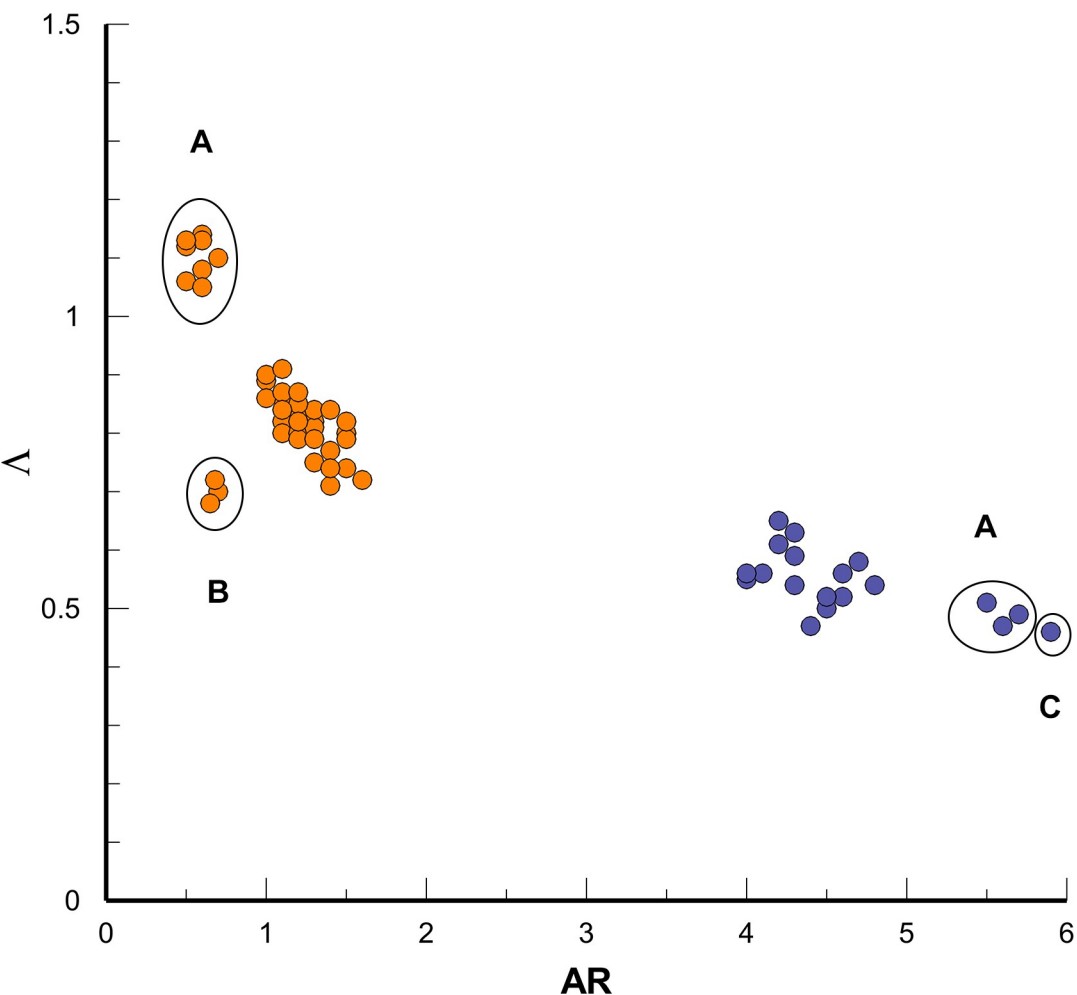

**Fig 18. The Λ, in radians vs *AR* scatterplot shows the fin planform variation in the dorsal fin (orange) and tail flukes (blue).** Different variation along the Λ and *AR* axes indicates two distinctive patterns of the fin planform. A–*G. melas*, B–*P. phocoena*, C–*B. acutorostrata*.

possess both the span-wise and chord-wise bending [55] that alters the thrust, lift and propulsive efficiency of the flapping foil [56,57]. The mode of flukes bending varies from the uniform bend of the entire fluke's blades to the characteristic shape of blended winglets of the airliners' wings [58,59]. The latter is related with a reduction in induced drag by decreasing the vortex formation on the wingtip area [59,60].

## Optimization of a generic shape to the specific function

The wing-like fin shape evolved in the interplay of the developmental, genetic, functional, and evolutionary factors as a result of a balance of four physical forces influencing a swimming or flying animal: Lift, drag, weight and thrust [61]. These forces act as the physical constraints that led to the appearance of a generic wing design with the swept-back tapered planform and foil-like cross-sectional geometry that maximize the *L/D* ratio [62,63]. In concert with a streamlined shape of the body, smooth skin and musculoskeletal system adaptations [64–67] it resulted in a dramatic improvement in speed, thrust production and efficiency in cetaceans compared with drag-based swimming in other taxa [4].

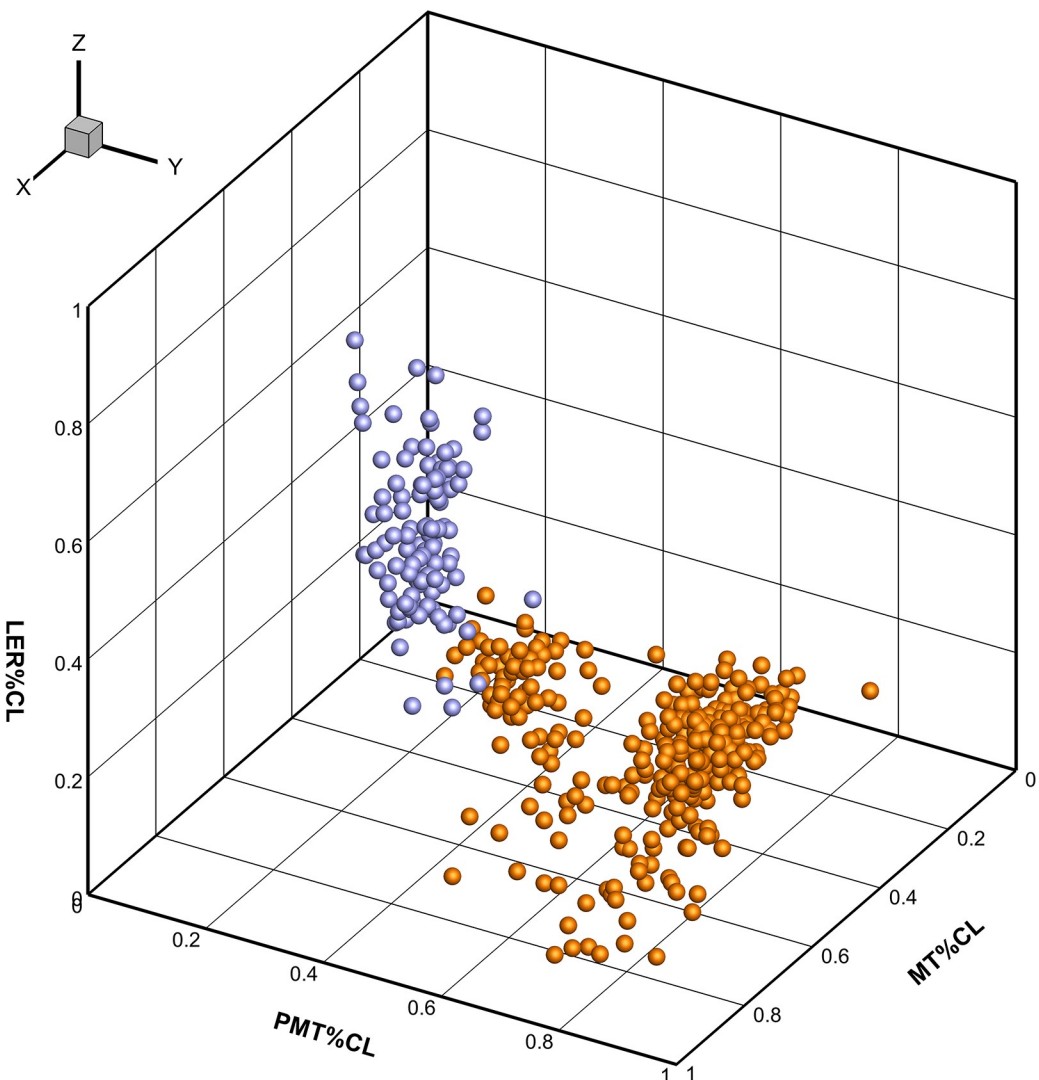

**Fig 19. Constraints in variation of cross-sections of the dorsal fin (orange) and tail flukes (blue) in the trait space of normalized non-dimensional parameters.** Different variation along the *LER%CL*, *PMT%CL* and *MT%CL* axes indicates two distinctive patterns of the cross-sectional geometry.

The divergence in a generic fin shape was found attributed to the primary function both on the planform and at the cross-sectional level. Distinctive patterns in the planform and cross-sectional geometry of the tail flukes appeared to be optimized for the flapping foil propulsion. With more variable planform, the cross-sectional geometry of the dorsal fin was similar to this one in the fixed vertical stabilizers in sailboats and aircrafts optimized for highest performance within a narrow range of α.

Constraints in variation of a generic wing-like shape, though, appear to be strong or weaken depending on the degree of specialization in a primary function. The tail flukes being the only organ of locomotion in cetaceans show a consistency in the wing planform, including an inverse relationship between Λ and *AR* across the representatives of both Odontoceti and Mysticeti [3]. Different combinations of low and high Λ and *AR* in tail flukes in cetaceans are related with lift production, reducing induced drag, and structural strength and stiffness of the fin tissues [14,55,68–70]. Dimensions of the tail flukes are related to the body length in

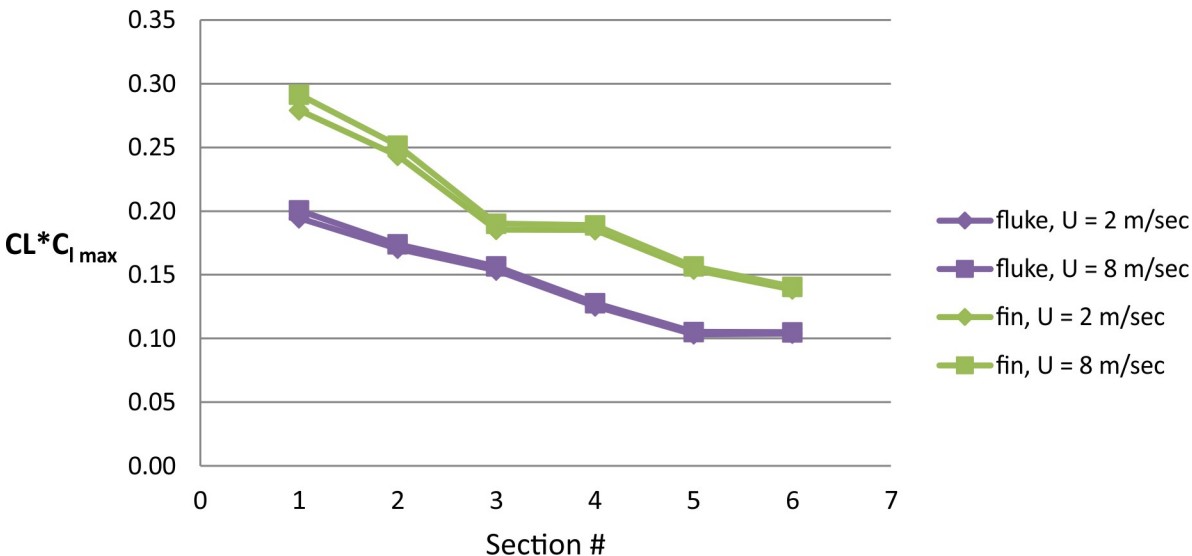

**Fig 20. Span-wise lift distribution.** Calculated for the cross-sections of the fluke (blue) and dorsal fin (green) of the *D. delphis* at simulated swimming speed 2 and 8 m/sec.

accordance with known empirical and theoretical scaling relationships between the body mass, fluke's planform, swimming kinematic parameters and swimming speed [28,55,71].

Apart from the flukes, the primary function of dorsal fin in stabilization, trim and maneuvering is shared between other appendages, tail peduncle and body [21,72]. The shape of the dorsal fin shows high interspecific variation where *S* and *A* of the dorsal fin vary considerably from a high *AR* fin in the killer whale *Orcinus orca*, to a low *AR* fin in *P. phocoena*, a dorsal ridge in river dolphins, and the absence of a fin in finless species such as the finless porpoise *Neophocaena phocaenoides* or Northern right whale dolphin *Lissodelphis borealis*. The spread out falcate shape of the dorsal fin can be observed across the different Odontoceti species but the contribution of fin to the stability and maneuverability control seems to be different in a 200 kg dolphin and a 7 ton bottlenose whale. On the intraspecific level, the distinct phenotypic variation of the dorsal fin was found in coastal and offshore ecotypes of dolphins [73–76].

The obtained results provide an insight into the evolutionary pathways of a generic fin shape driven by specialization in a primary function. Further studies of the developmental, genetic, and environmental drivers of fins variation would be the key in understanding the mechanisms of shaping the performance envelope of species having different habitat preference and feeding specialization.

This work could serve as a starting point in further studies of the effect of span-wise and chord-wise bending of the bio-inspired flapping foils in wake formation and thrust generation. It could also inspire the development of propulsors and control surfaces for AUV's operating within the similar range of Reynolds numbers $10^5$–$10^7$. Additionally, it could help in optimization of the external design of the fin-mounted tags for small cetacean telemetry.

## Supporting information

**S1 Fig. Span-wise distribution of the chord length CL of the dorsal fin cross-sections, means ± SD.**
(TIF)

**S2 Fig. Span-wise distribution of the maximal thickness MT of the dorsal fin cross-sections, means ± SD.**
(TIF)

**S3 Fig. Span-wise distribution of the position of maximal thickness PMT of the dorsal fin cross-sections, means ± SD.**
(TIF)

**S4 Fig. Span-wise distribution of the relative maximal thickness MT, %CL of the dorsal fin cross-sections, means ± SD.**
(TIF)

**S5 Fig. Span-wise distribution of the relative position of maximal thickness PMT, %CL of the dorsal fin cross-sections, means ± SD.**
(TIF)

**S6 Fig. Span-wise distribution of the relative leading edge radius LER, %CL of the dorsal fin cross-sections, means ± SD.**
(TIF)

**S7 Fig. Span-wise distribution of the chord length CL of the tail flukes cross-sections, means ± SD.**
(TIF)

**S8 Fig. Span-wise distribution of the maximal thickness MT of the tail flukes cross-sections, means ± SD.**
(TIF)

**S9 Fig. Span-wise distribution of the position of maximal thickness PMT of the tail fluke cross-sections, means ± SD.**
(TIF)

**S10 Fig. Span-wise distribution of the relative maximal thickness MT, %CL of the tail fluke cross-sections, means ± SD.**
(TIF)

**S11 Fig. Span-wise distribution of the relative position of maximal thickness PMT, %CL of the tail fluke cross-sections, means ± SD.**
(TIF)

**S12 Fig. Span-wise distribution of the relative leading edge radius LER, %CL of the tail flukes cross-sections, means ± SD.**
(TIF)

**S13 Fig. Span-wise distribution of the drag coefficient of the dorsal fin cross-sections calculated at 2 m/sec, means ± SD.**
(TIF)

**S14 Fig. Span-wise distribution of the drag coefficient of the dorsal fin cross-sections calculated at 8 m/sec, means ± SD.**
(TIF)

**S15 Fig. Span-wise distribution of the drag coefficient of the tail flukes cross-sections calculated at 2 m/sec, means ± SD.**
(TIF)

**S16 Fig. Span-wise distribution of the drag coefficient of the tail flukes cross-sections calculated at 8 m/sec, means ± SD.**
(TIF)

**S1 File. IUCN status and coordinates.**
(XLSX)

**S2 File. Cross-sectional parameters.**
(XLSX)

**S3 File. Fin normalized coordinates.**
(XLSX)

**S4 File. Fin sections coefficients, U = 2 msec.**
(XLSX)

**S5 File. Fin sections coefficients, U = 8 msec.**
(XLSX)

**S6 File. Flukes normalized coordinates.**
(XLSX)

**S7 File. Fluke sections coefficients, U = 2 msec.**
(XLSX)

**S8 File. Fluke sections coefficients, U = 8 msec.**
(XLSX)

## Acknowledgments

The authors thank to the collaborators of the LIENSS Institute of the University of La Rochelle, L'Observatoire PELAGIS UMS 3462, Museum of Natural History of the Faroe Islands, and Institute for Terrestrial and Aquatic Wildlife Research (ITAW) of the University of Veterinary Medicine Hannover, Foundation for their help and huge efforts in handling and taking morphometric measurements of cetaceans.

## Author Contributions

**Conceptualization:** Vadim Pavlov.

**Formal analysis:** Vadim Pavlov.

**Funding acquisition:** Cecile Vincent, Ursula Siebert.

**Investigation:** Vadim Pavlov, Cecile Vincent, Bjarni Mikkelsen, Justine Lebeau.

**Project administration:** Vincent Ridoux.

**Supervision:** Ursula Siebert.

**Writing – original draft:** Vadim Pavlov.

**Writing – review & editing:** Vadim Pavlov, Cecile Vincent, Bjarni Mikkelsen, Justine Lebeau, Vincent Ridoux, Ursula Siebert.

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
