## [Decision Letter · Decision Letter 0]

7 Oct 2020

PONE-D-20-22457

Form, function, and divergence of a generic fin shape in small cetaceans

PLOS ONE

Dear Dr. Pavlov,

Thank you for submitting your manuscript to PLOS ONE. After careful consideration, we feel that it has merit but does not fully meet PLOS ONE’s publication criteria as it currently stands. Therefore, we invite you to submit a revised version of the manuscript that addresses the points raised during the review process.

We look forward to receiving your revised manuscript.

Kind regards,

Songhai Li, Ph.D.

Academic Editor

PLOS ONE

Journal Requirements:

2. (1) Please describe how you went about collecting the various specimens, that is: were you notified by various entities, individuals prior to obtaining them? (2) Indicate whether the specimens that were bycatch had been caught by commercial fishermen. (3) Please provide a supplemental table that specifies the geographic coordinates for each of the specimens you collected for this research. (4) Additionally, please indicate whether you obtained any type of permit/authorization from any local or regional agency for collecting the specimens. (5) Finally, please specify the IUCN Red List status for each of the specimens collection: https://www.iucnredlist.org/

"The funders had no role in study design, data collection and analysis, decision to publish, or preparation of the manuscript.".

Reviewers' comments:

Reviewer's Responses to Questions

**Comments to the Author**

1. Is the manuscript technically sound, and do the data support the conclusions?

Reviewer #1: Partly

Reviewer #2: Partly

Reviewer #3: Partly

2. Has the statistical analysis been performed appropriately and rigorously? 

Reviewer #1: I Don't Know

Reviewer #2: No

Reviewer #3: No

3. Have the authors made all data underlying the findings in their manuscript fully available?

Reviewer #1: Yes

Reviewer #2: Yes

Reviewer #3: No

4. Is the manuscript presented in an intelligible fashion and written in standard English?

Reviewer #1: Yes

Reviewer #2: Yes

Reviewer #3: No

5. Review Comments to the Author

Reviewer #1: Overall, I think there are interesting ideas being touched upon in this manuscript but they are lost in the papers current structure. The components are here but there is a significant amount of work needed to make them cohesive. I recommend reorganizing the figures as several key anatomical figures are not discussed until the end of the paper and some data figures are not included in the main text. Additionally, all of the sections need some work to be made into a more cohesive package, The introduction set’s up a background for the paper but doesn’t adequately introduce what’s being done/analyzed and why. The results section needs more detail in a couple of areas and lacks several explanations which are only brought up later in the materials and methods. Finally, the discussion section feels disconnected and largely unrelated to the analyses that were highlighted in the results section. There is little discussion of the numeric results or the CFD components and the geometric aspects of the work are discussed in only vague or general terms without specific references to the study taxa and the measurements and analyses done there. For example, Figure 3, which is the most data dense figure, is not mentioned in the discussion in any way despite this being the seeming center point of their geometric comparisons.

General Comments

1. Sample sizes need to be reported earlier in the document and in the tables. Not just at the end of the document. These sample sizes are relevant to some of the results and discussion but they are not addressed

2. Explanations need to be given for why sub-sets of taxa are analyzed with CFD

3. With a few early exceptions very few abbreviations need to be defined when they first appear in the manuscript. Most are defined much later in the manuscript (materials and methods)

4. Genus Species combinations need to be italicized in tables and elsewhere. I would also suggest referring to them this way throughout the manuscript rather than using common names in the main text (or including the common names in your tables). The current usage makes it much harder to follow what’s going on in the main text for anyone who does not frequently work with these taxa.

5. Grammar should be revisited as a few sentences have confusing structure and grammar.

Specific Comments

1. Citation at line 62 feels unnecessary, as is the quotation.

2. Line 71-77 This paragraph need to be restructured. It currently feels disorganized and it’s point is unclear.

3. Line 72-73 I’m not entirely certain what this sentence is meant to convey. Is it a general statement like “Appendages with this morphology typically generate high lift forces and low drag forces”

4. Line 74-75 is missing a citation

5. Line 76-77 This sentence feels incomplete as written. I’m not able to tell if the intent is to simply highlight that these relationships change throughout the animals life or if these changes signal differences in swimming habits through ontogeny.

6. Line 79: provide a range of aspect ratios here for context.

7. Line 104: Define CI, you have not yet done so.

8. Line 126: You’ve not defined these abbreviations

9. Line 144-146: I disagree with this result. These two species are “less variable purely because they have the lowest sample size which has not been accounted for.

10. Line 162: I would like to see a table of some of these values either in supplement or in the main text. The supplemental figures are difficult to read and exact values can’t be seen. Also by what metric is the variance in these parameters being declared “insignificant”? keeping in mind the variance here is likely affected by sample size which hasn’t been accounted for

11. Line 205: I wasn’t sure what was meant here

12. Line 215: this seems like a significant difference. which species had the short transition zone? Which had the longest? Add more details to this section don’t rely solely on supplement.

13. Line 256: I’m singling this out but it is relative to other sections of the document. Be more specific about the parameters you’re referring to. Rather than saying “hydrodynamic performance”. Talk about the specific changes in parameters such as drag coefficient or lift. These are significantly less vague and will help a reader understand how you reach your conclusions.

14. Line 272: Avoid the use of best here, instead, as was done earlier use highest or lowest

15. Line 280: I find it somewhat disconcerting that you refer to lift production here but you haven’t really discussed lift nor presented any lift data up to this point in the document. If you intend for the discussion of pressure to be synonymous with lift that needs to be more clear

16. Line 372-373: The connection to figure 1 here needs to be discussed more

17. Line 382-387: There should be more discussion of your numerical results throughout the discussion section or at least references back to your numerical results when discussing generally. Right now there is very little connection between the points in the discussion on how the numbers you get in your results substantiate this. There are only two figure references in this section 1 supplemental and 1 main body. The rest of the section vaguely discusses general trends and comparisons without mentioning the data in any direct way.

18. Line 454-478: What were the settings used for the fluid? I’m not familiar with this particular CFD package but if it has presets for water then it should at least be stated that those were used.

19. Line 472-478: Why were these particular animals the only ones analyzed to this level of detail?

Figure and tables:

1. Axes in all main text figures need labels. In my opinion at least some amount of the data that is in the supplemental figures also needs to be moved to the main body figures. A great deal of the referenced numbers/results are in supplement and lack any representation in the main text which will make reading and interpreting the paper without the supplement exceptionally difficult

2. Figures 4, 5, 6, and 7 appear far too late in the document to be helpful to a reader. At the very least their reference needs to be moved up in the text to perform there introductory functions.

3. Table 3: why is B. acutorostrata absent from this table?

4. Table 4: Why are only 3 species being used here? How were they selected? This should at least be discussed and explained in the material and methods section.

Reviewer #2: This study examines the structure of the caudal flukes and dorsal fin in small cetaceans from a morphological and hydrodynamic perspective. The authors examined the planform and cross-sectional geometry of these derived structures from a variety of dolphins and porpoise. The information was then analyzed with multivariate statistics and hydrodynamics. The data presented helps to expand the already existing data on fluke and fin geometry and hydrodynamics. The manuscript, however, has a number of limitations with its presentation. First, there are no hypotheses presented in the Introduction. What was being tested? Bose et al. (1990) and Fish and Rohr (1999) present data on the geometry of flukes and fins. A case should be made for how the present study expands on the knowledge base. In addition, past studies on the structure of the flukes and dorsal fin should be integrated into the text, including:

Felts, W. J. L. (1966). Some functional and structural characteristics of cetacean flippers and flukes. In K. S. Norris (ed.): Whales, Dolphins, and Porpoises. University of California Press: Berkeley. pp. 255-276.

Gough, W. T., Fish, F. E., Wainwright, D. K. and Bart-Smith, H. (2018). Morphology of the core fibrous layer of the cetacean tail fluke. Journal of Morphology 279(6): 757-765.

The internal structure is a novel design feature shared by both the flukes and dorsal fin. Both lack bony skeletal elements for support and alternatively have an internal arrangement of a dense array of collagen fibers. The stiffness supplied by the collagen fibers is important in both structures. The dorsal fin is only a control surface that is oriented in the vertical plane to aid in stabilizing the body in straight-line swimming and during turning. The flukes can act as a control surface but are in the horizontal plane oriented 90 degrees from the orientation of the dorsal fin. A direct statistical comparison of the two control surfaces would be advisable. Particularly as any performance and structural differences could be affect by the position on the body with the dorsal fin near the center of gravity and the flukes farthest away. In addition being composed of the same materials, the flukes have two other functions could be related to differences in structure and hydrodynamics, namely the flukes can be twisted for turning (see Fish, 2002) and of course are oscillated for propulsion. For this later point, flexibility plays an important part of the functioning of the flukes, which are bent in both the chordwise and spanwise directions. Chordwise bending of the flukes for a swimming dolphin is shown in Romanenko EV (2002) Fish and Dolphin Swimming (Pensoft, Sofia) p 429. The spanwise bending at the fluke tips to act as winglets is provided in Fish and Lauder (2017). The bending may also be expressed as differences in 3D geometry between the flukes and dorsal fin. In discussing the hydrodynamics of the flukes, another reference to discuss is:

Ayancik, F., Fish, F. E. and Moored, K. W. 2020. Three-dimensional scaling laws of cetacean propulsion characterize the hydrodynamic interplay of flukes’ shape and gait. Journal of the Royal Society Interface 17:20190655.

Another limitation of the manuscript is a general lack of detail in the Methods and discussion of the Results and throughout the Discussion. What statistical tests were used to detect differences between species? And if not, why not? The only statistical test appears to be Principle Component Analysis. How was the PCA performed? What variables were used? Although PCA data are displayed, there is little comment on the results. What do axes F1 and F2 represent? This analysis does not appear to be discussed in the Discussion. How did size of the various cetaceans affect the results? For the hydrodynamic analysis there is no indication if angle of attack was varied. The polar plots indicate this, but there is no indication of this in the Methods. The panel method has limitations with respect to high angles of attack. How was this dealt with? A more detailed and discussion is necessary.

Reviewer #3: This study describes the shape of dorsal fins and tail flukes in cetaceans of a range of body sizes, and interprets the observed variation with respect to fluid dynamics.

The study presents results of primary scientific research; the study consists of measurements of the morphology of multiple animals and comparisons across species, and assessment of hydrodynamic performance of dorsal fins and flukes at two swimming speeds in comparison with standard airfoils.

These results reported have not been published elsewhere, as far as I can tell.

The detail provided concerning the experiments, statistics, and other analyses could be further expanded to aid readers in understanding of the study, and elements of the technical excellence should be improved before the paper is accepted for publication. I highlight several specific issues below.

In terms of presentation, it would be imperative to provide readers with information concerning the study sample size, which I was not able to determine. Also, although the authors say that the species are listed in Table 1 in order of size, it would be very helpful to give the “size” of each specimen or species to let readers understand this aspect of the study. Both of these could be addressed by expanding Table 1. In general, readers would benefit greatly from much more informative figure captions.

I think that the role of size as a design factor could be brought out more effectively overall. The authors note in the Introduction that “ Larger wings generate more lift than wings of smaller area or span and are associated with increased thrust production and stability control of large species when swimming. The relationships between S, A, and BL is different in life history stages that affect the swimming performance of cetaceans (lines 74 – 77).” But scaling principles aren’t employed to make baseline predictions about patterns of variation of various traits with respect to body size. This would make the study much more powerful. For example, the study finds that “absolute” parameters differ more cross the sample than do “relative” ones (which should be defined); surely this would be what one predicts? Based on scaling alone, one would predict that “relative” variables, which are ratios or angles, would tend not to vary with animal size. Because this idea isn’t developed in the Introduction, or brought up in the Discussion, readers are never able to appreciate the significance of any variation, or its absence, across the size sample.

There is an additional aspect of study design related to body size that requires further attention in the text, at the very least. All hydrodynamic performance simulations were carried out at two flow speeds, 2 and 8 m/s, which are categorized as cruising and burst speed. However, no biological rationale is provided for the selection of the speeds as cruising vs. burst, and most importantly, the analysis assumes that a given cruising or burst speed is appropriate for animals that vary many-fold in body size. This assumes that 2 m/s, or 8 m/s, is the same for a 75 kg harbor porpoise as for a 2 or 3 metric ton minke whale. This assumption seems wrong. Larger animals move more quickly, so a comparable speed would be one based on a certain number of body lengths per second or Strouhal number. The authors must explain their approach to this issue and carefully justify it. The rest of the study is difficult to interpret in advance of resolving this issue, because comparisons of results of the smallest and largest species at the same absolute speed does not make biological or physical sense.

Also, if a primary goal of the study, as suggested by the first paragraph, is to provide insight into evolutionary pathways for divergence of a generic shape driven by the distinct roles of the dorsal fins and flukes in stability control and thrust production, it would be helpful to readers if the authors could propose hypotheses about how these should differ between the two sets of structures, and indicate how their analyses will test these specific hypotheses. There is some general language that describes differences in structure and function of dorsal fins and tail flukes, but this is not translated into specific predictions for hydrodynamic characteristics, the primary subject of the analysis. If dorsal fins are single fixed wings with stability as a primary function, and tail flukes are paired oscillating wings that function to provide control and for which increased lift and delayed stall are critical (please explain why), then what do you predict the simulations might discover as key differences between them? How should this be affected by body size and flow speed? There is one comparison (Fig S25) that is said to demonstrate critical differences between dorsal fins and flukes, but they come from different species (common dolphin fluke and Atlantic white-beaked dolphin dorsal fin), which seems highly problematic. Is this the only analysis that is relevant?

The other major arena in which the analysis is weak concerns the phylogenetic relationships among the species in the sample. The species included in this study are not independent data points. Five of the seven study species, D. delphis, L. acutus, L. albirostris, T. truncatus, and G. melas, are members of a single family, the Delphinidae. P. phocoena is a member of the sister taxon, family Phocoenidae. B. acutorostrata is a member of not only a distinct family, Balaenidae, this family is in a completely different part of the cetacean suborder, the Mysticeti or baleen, rather than Odontoceti or toothed whales. So, the degree of evolutionary relationship between species pairs varies greatly. I don’t mean to say that the authors need to conduct a full, phylogenetically-informed analysis. But, particularly if they wish to cast the study in an evolutionary light, they must acknowledge the evolutionary relationships among the taxa. The study species differ in body size, ecology, and phylogeny. Attributing observed differences to a single factor is therefore impossible, and interpretations must proceed with caution.

Finally, any time simulation approaches are adopted, particularly for biological applications, it is valuable to provide the reader with validation of some kind. How do the results of the code employed by the authors, DesignFOIL™, compare to results of empirical studies of the same phenomena? What is lost by carrying out analysis in 2D? I have no problem with the choice of 2D analysis, but this should be justified to the reader, and the paper should provide readers with concrete understanding of the potential shortcomings or errors produced by the simulation approach. No simulation is perfect, so we should always treat simulation in a manner that identifies the results in which we have greatest confidence, and those that are less reliable. The authors need to be clearer about the strengths but also weaknesses of their simulations. Some of the results provided from simulations (Suppementary figures S1-20) have some type of error bar, others (S21-25) have none. Is this due only to the number of specimens? If so, does this mean that no error estimates are included? How are we to interpret this – with no error estimates, it is difficult to understand how to say whether two species are similar or different.

The paper’s conclusions are largely reasonable, but in are not always supported by the data. One striking shortcoming is that although hydrodynamics is a/the central focus of the study, none of the paper’s main figures present any results of the hydrodynamics investigations; these are reserved solely for the Supplemental Information. This does not make sense to me. If the authors think that the most important results concern some of the CFD analyses, they should identify these and select or prepare figures that convey these results.

The article is presented in a mostly intelligible fashion and is largely written in standard English. However, there are numerous places where clarity and precision could be improved through editing by an expert or native speaker of English.

The research meets all applicable standards for the ethics of experimentation and research integrity.

I do not believe that the article adheres to appropriate reporting guidelines and community standards for data availability. The authors state that “All relevant data are within the manuscript and its Supporting Information files”, but this does not appear to be true. The results are available in the manuscript and supporting information, but not the underlying data, and perhaps there is some confusion on the part of the authors concerning data availability. Community members could view plots of the output from simulations of flow around cross-sections of dorsal fins and flukes, for example, but neither the outlines of the hydrofoils nor the raw outputs of the simulations are included as data files.

6. PLOS authors have the option to publish the peer review history of their article (what does this mean?). If published, this will include your full peer review and any attached files.

Reviewer #1: No

Reviewer #2: No

Reviewer #3: No

---

## [Author Response · Author response to Decision Letter 0]

17 Mar 2021

Reviewers' comments:

Reviewer's Responses to Questions

Comments to the Author

1. Is the manuscript technically sound, and do the data support the conclusions?

Reviewer #1: Partly

Reviewer #2: Partly

Reviewer #3: Partly

2. Has the statistical analysis been performed appropriately and rigorously? 

Reviewer #1: I Don't Know

Reviewer #2: No

Reviewer #3: No

3. Have the authors made all data underlying the findings in their manuscript fully available?

Reviewer #1: Yes

Reviewer #2: Yes

Reviewer #3: No

4. Is the manuscript presented in an intelligible fashion and written in standard English?

Reviewer #1: Yes

Reviewer #2: Yes

Reviewer #3: No

5. Review Comments to the Author

Dear Editor and Reviewers,

The authors are grateful for the valuable comments and suggestions that resulted in a number of the improvements and corrections over the manuscript. We appreciate it a lot. Please find our comments below.

Reviewer #1: Overall, I think there are interesting ideas being touched upon in this manuscript but they are lost in the papers current structure. The components are here but there is a significant amount of work needed to make them cohesive. I recommend reorganizing the figures as several key anatomical figures are not discussed until the end of the paper and some data figures are not included in the main text. 

- The structure of the manuscript is reorganized now with the Materials and Methods section following the Introduction. New pictures are added and the pictures are reorganized to give the reader a clear idea about key anatomical features.

Additionally, all of the sections need some work to be made into a more cohesive package, The introduction set’s up a background for the paper but doesn’t adequately introduce what’s being done/analyzed and why. 

- The Introduction is changed now with our hypotheses included to help better introduce what’s being done, Page 3, Line 92-98.

The results section needs more detail in a couple of areas and lacks several explanations which are only brought up later in the materials and methods. 

- The reorganized Materials and Methods section is ahead the Results now. Table 1 and additional details are added to the “Hydrodynamic characteristics of the cross-sections of fins” subsection of the Materials and Methods, Page 7. The “Statistical analysis” subsection is also added to the Materials and Methods, Page 7, Line 234-241. 

- New Tables 3 and 5, and new Figures 7,9,11 describing the shape of the fins and fins cross-sections are added to the Results.

- ANOVA tables are added to the Results, Page 11-13, Tables 6-8.

- New Figures 12-17, and 18-19 are added to the “Comparison of the shape and cross-sections of the dorsal fin and tail flukes” subsection of the Results, Page 18, Line 486-493. 

Finally, the discussion section feels disconnected and largely unrelated to the analyses that were highlighted in the results section. There is little discussion of the numeric results or the CFD components and the geometric aspects of the work are discussed in only vague or general terms without specific references to the study taxa and the measurements and analyses done there. For example, Figure 3, which is the most data dense figure, is not mentioned in the discussion in any way despite this being the seeming center point of their geometric comparisons.

- The Discussion section is changed considerably and better linked to the results now. The geometry and hydrodynamic characteristics of the fins and fin cross-sections are described in terms of the wing and airfoil parameters and hydrodynamic coefficients. Figure 3 (now Fig 10) and twelve additional figures prepared for this version of the manuscript are discussed in this section. 

General Comments

1. Sample sizes need to be reported earlier in the document and in the tables. Not just at the end of the document. These sample sizes are relevant to some of the results and discussion but they are not addressed

- Sample size now is reported in the Materials and Methods section following the Introduction, Page 4, Line 129, 143-144.

2. Explanations need to be given for why sub-sets of taxa are analyzed with CFD

- All studied species are analyzed now.

3. With a few early exceptions very few abbreviations need to be defined when they first appear in the manuscript. Most are defined much later in the manuscript (materials and methods)

- Done, Page 4-5, Line 167-187.

4. Genus Species combinations need to be italicized in tables and elsewhere. I would also suggest referring to them this way throughout the manuscript rather than using common names in the main text (or including the common names in your tables). The current usage makes it much harder to follow what’s going on in the main text for anyone who does not frequently work with these taxa. 

- Genus Species combinations are italicized and common names are removed elsewhere in the manuscript. 

5. Grammar should be revisited as a few sentences have confusing structure and grammar.

- Done.

Specific Comments

1. Citation at line 62 feels unnecessary, as is the quotation. 

- This citation is removed, Page 2, Line 63.

2. Line 71-77 This paragraph need to be restructured. It currently feels disorganized and it’s point is unclear. 

3. Line 72-73 I’m not entirely certain what this sentence is meant to convey. Is it a general statement like “Appendages with this morphology typically generate high lift forces and low drag forces”

4. Line 74-75 is missing a citation

5. Line 76-77 This sentence feels incomplete as written. I’m not able to tell if the intent is to simply highlight that these relationships change throughout the animals life or if these changes signal differences in swimming habits through ontogeny.

- This paragraph is changed considerably and missing citation of the different patterns of swimming in calves and adult animals is added, Page 2-3, Line 73-76.

6. Line 79: provide a range of aspect ratios here for context. 

- Done. Page 2, Line 77-79.

7. Line 104: Define CI, you have not yet done so.

- Done. Page 5, Line 177.

8. Line 126: You’ve not defined these abbreviations

- Done. Page 5, Line 185-187.

9. Line 144-146: I disagree with this result. These two species are “less variable purely because they have the lowest sample size which has not been accounted for.

- This statement is removed, Page 9, Line 295.

10. Line 162: I would like to see a table of some of these values either in supplement or in the main text. The supplemental figures are difficult to read and exact values can’t be seen. Also by what metric is the variance in these parameters being declared “insignificant”? keeping in mind the variance here is likely affected by sample size which hasn’t been accounted for

- The measurements data are added to the supplement (S2 File. Cross-sectional parameters.xlsx). This statement is removed from the text.

11. Line 205: I wasn’t sure what was meant here

- Now it is replaced by the “Regarding peculiarity of pressure distribution, there was an absence of zero or a slightly positive pressure gradient”, Page 14, Line 397-399.

12. Line 215: this seems like a significant difference. which species had the short transition zone? Which had the longest? Add more details to this section don’t rely solely on supplement.

- Discussion of the length of the laminar and transition zone is removed from the manuscript. The number of the parameters is reduced to the hydrodynamic coefficients now.

13. Line 256: I’m singling this out but it is relative to other sections of the document. Be more specific about the parameters you’re referring to. Rather than saying “hydrodynamic performance”. Talk about the specific changes in parameters such as drag coefficient or lift. These are significantly less vague and will help a reader understand how you reach your conclusions.

- This section is changed considerably now and the “hydrodynamic performance” is removed from this section, Page 17-18, Line 465-485. 

14. Line 272: Avoid the use of best here, instead, as was done earlier use highest or lowest

- Done. Page 17, Line 478-479.

15. Line 280: I find it somewhat disconcerting that you refer to lift production here but you haven’t really discussed lift nor presented any lift data up to this point in the document. If you intend for the discussion of pressure to be synonymous with lift that needs to be more clear

- The Discussion section is changed considerably and includes the discussion of the hydrodynamic characteristics of the cross-sections added to the Results section in terms of hydrodynamic coefficients, Page 18-20, Line 507-573. 

16. Line 372-373: The connection to figure 1 here needs to be discussed more

- The reference to Fig 1 is removed from this sentence, Page 21, Line 621.

17. Line 382-387: There should be more discussion of your numerical results throughout the discussion section or at least references back to your numerical results when discussing generally. Right now there is very little connection between the points in the discussion on how the numbers you get in your results substantiate this. There are only two figure references in this section 1 supplemental and 1 main body. The rest of the section vaguely discusses general trends and comparisons without mentioning the data in any direct way.

- More details are added to the discussion of the CFD in this section with the references to the thirteen figures and four tables, Page 18-20, Line 507-573.

18. Line 454-478: What were the settings used for the fluid? I’m not familiar with this particular CFD package but if it has presets for water then it should at least be stated that those were used.

- The DesignFoil software utilizes atmospheric and performance parameters (density, pressure, temperature, speed of sound, air viscosity, etc.) for calculating the Re numbers to test the airfoils. In our study we calculated the appropriate Re numbers based on the fresh water viscosity, length of a cross-section and two swimming speeds, two m/sec and eight m/sec respectively. 

19. Line 472-478: Why were these particular animals the only ones analyzed to this level of detail?

- This type of analysis was initially added to the manuscript as an illustration of the span-wise lift distribution on the dorsal fin and tail flukes. Now it represented on the dorsal fin and tail flukes of the same species, D. delphis, as it was also mentioned by the Reviewer 3, Page 18, Line 483-485.

Figure and tables:

1. Axes in all main text figures need labels. In my opinion at least some amount of the data that is in the supplemental figures also needs to be moved to the main body figures. A great deal of the referenced numbers/results are in supplement and lack any representation in the main text which will make reading and interpreting the paper without the supplement exceptionally difficult

- Labels are added to the axes in all main text figures.

- Twelve additional figures and three tables are added now to represent the results in the main text.

2. Figures 4, 5, 6, and 7 appear far too late in the document to be helpful to a reader. At the very least their reference needs to be moved up in the text to perform there introductory functions.

- The manuscript structure is reorganized now and the additional figures are added. Figures 4, 5, 6, and 7 are the now the Figures 1-5 in the Materials and Methods section, Page 4-7.

3. Table 3: why is B. acutorostrata absent from this table?

- The B. acutorostrata data are added now to this table, Page 15, Table 9.

4. Table 4: Why are only 3 species being used here? How were they selected? This should at least be discussed and explained in the material and methods section.

- Missing species are added to this table, Page 16-17, Table 10.

Reviewer #2: This study examines the structure of the caudal flukes and dorsal fin in small cetaceans from a morphological and hydrodynamic perspective. The authors examined the planform and cross-sectional geometry of these derived structures from a variety of dolphins and porpoise. The information was then analyzed with multivariate statistics and hydrodynamics. The data presented helps to expand the already existing data on fluke and fin geometry and hydrodynamics. The manuscript, however, has a number of limitations with its presentation. First, there are no hypotheses presented in the Introduction. What was being tested? Bose et al. (1990) and Fish and Rohr (1999) present data on the geometry of flukes and fins. A case should be made for how the present study expands on the knowledge base. In addition, past studies on the structure of the flukes and dorsal fin should be integrated into the text, including:

- Two hypotheses about a generic fin shape in cetaceans are added to the Introduction section, Page 3, Line 92-98.

- Although the geometry of the flukes and fins is presented in Bose et al. (1990), Fish and Rohr (1999) and in other publications, this paper presents the first detailed analysis of the cross-sectional geometry across the selected species and two types of fins.

Felts, W. J. L. (1966). Some functional and structural characteristics of cetacean flippers and flukes. In K. S. Norris (ed.): Whales, Dolphins, and Porpoises. University of California Press: Berkeley. pp. 255-276.

Gough, W. T., Fish, F. E., Wainwright, D. K. and Bart-Smith, H. (2018). Morphology of the core fibrous layer of the cetacean tail fluke. Journal of Morphology 279(6): 757-765. 

- Our paper is focused on the analysis of geometry of fins and fin cross-sections and hydrodynamic characteristics of the fin cross-sections. We integrated the study of Gough et al 2018 in the discussion Page 21, Line 599, but overall, the morphology and histology of the fin tissues are beyond the focus of this study. 

The internal structure is a novel design feature shared by both the flukes and dorsal fin. Both lack bony skeletal elements for support and alternatively have an internal arrangement of a dense array of collagen fibers. The stiffness supplied by the collagen fibers is important in both structures. The dorsal fin is only a control surface that is oriented in the vertical plane to aid in stabilizing the body in straight-line swimming and during turning. The flukes can act as a control surface but are in the horizontal plane oriented 90 degrees from the orientation of the dorsal fin. A direct statistical comparison of the two control surfaces would be advisable. 

- We added ANOVA to verify our hypotheses about a generic fin shape and to examine the linkage of the cross-sectional geometry variation to the species, fin types, and length of the body, Page 11-13, Tables 6-8. 

- Two tables illustrating correlations of the wing parameters of the dorsal fin and tail flukes are also added, Page 8, Table 3 and Page 10, Table 4. 

- We added the � vs A scatterplot illustrating the variation of both types of fins in species studied, Page 18, Fig.18, Line 486-488.

Particularly as any performance and structural differences could be affect by the position on the body with the dorsal fin near the center of gravity and the flukes farthest away. In addition being composed of the same materials, the flukes have two other functions could be related to differences in structure and hydrodynamics, namely the flukes can be twisted for turning (see Fish, 2002) and of course are oscillated for propulsion. 

- The idea that both the dorsal fin and tail flukes serve several functions is discussed in the Discussion section, Page 20, Line 560-573. 

For this later point, flexibility plays an important part of the functioning of the flukes, which are bent in both the chordwise and spanwise directions. Chordwise bending of the flukes for a swimming dolphin is shown in Romanenko EV (2002) Fish and Dolphin Swimming (Pensoft, Sofia) p 429. The spanwise bending at the fluke tips to act as winglets is provided in Fish and Lauder (2017). -it is already discussed in the text. The bending may also be expressed as differences in 3D geometry between the flukes and dorsal fin. In discussing the hydrodynamics of the flukes, another reference to discuss is:

- Both the spanwise and chordwise bending influencing the forces generated by fins is discussed, Page 20, Line 568-573. 

- Papers by Romanenko EV (2002) and Ayancik et al 2020 are added in the Discussion, Page 20, Line 569, 589.

Ayancik, F., Fish, F. E. and Moored, K. W. 2020. Three-dimensional scaling laws of cetacean propulsion characterize the hydrodynamic interplay of flukes’ shape and gait. Journal of the Royal Society Interface 17:20190655.

Another limitation of the manuscript is a general lack of detail in the Methods and discussion of the Results and throughout the Discussion. What statistical tests were used to detect differences between species? And if not, why not? The only statistical test appears to be Principle Component Analysis. How was the PCA performed? What variables were used? Although PCA data are displayed, there is little comment on the results. What do axes F1 and F2 represent? This analysis does not appear to be discussed in the Discussion. How did size of the various cetaceans affect the results? For the hydrodynamic analysis there is no indication if angle of attack was varied. The polar plots indicate this, but there is no indication of this in the Methods. The panel method has limitations with respect to high angles of attack. How was this dealt with? A more detailed and discussion is necessary.

- ANOVA was added to verify our hypotheses about a generic fin shape in cetaceans and to examine the linkage of the cross-sectional geometry variation to the species, fin types, and length of the body, Page 11-13, Tables 6-8. 

- We used three dimensionless parameters of the cross-sections as PCA variables to study the patterns of cross-sectional geometry variation. Axes 1 and 2 represented the shape of the foil (cross-section) and the foil thickness respectively, Page 13, Line 361-373. 

- The angles of attack are indicated in the Materials and Methods section, Page 7, Line 225-230. We did not use the angles of attack exceeding 20 degrees in our study. 

Reviewer #3: This study describes the shape of dorsal fins and tail flukes in cetaceans of a range of body sizes, and interprets the observed variation with respect to fluid dynamics.

The study presents results of primary scientific research; the study consists of measurements of the morphology of multiple animals and comparisons across species, and assessment of hydrodynamic performance of dorsal fins and flukes at two swimming speeds in comparison with standard airfoils.

These results reported have not been published elsewhere, as far as I can tell.

The detail provided concerning the experiments, statistics, and other analyses could be further expanded to aid readers in understanding of the study, and elements of the technical excellence should be improved before the paper is accepted for publication. I highlight several specific issues below.

- The manuscript is changed considerably to help readers in understanding of the study. The main text structure is reorganized now with the Materials and Method section ahead of the Results. Our hypotheses about a generic fin shape in cetaceans and the linkage of the cross-sectional geometry variation to the species, fin types, and length of the body were added to the Introduction, Page 3, Line 92-98. 

- ANOVA was added to verify our hypotheses and the results are discussed in more details in terms of the wing and airfoil parameters and hydrodynamic coefficients, Page 11-13, Tables 6-8. 

- Twelve additional figures and two tables are added now to represent the results in the main text.

In terms of presentation, it would be imperative to provide readers with information concerning the study sample size, which I was not able to determine. Also, although the authors say that the species are listed in Table 1 in order of size, it would be very helpful to give the “size” of each specimen or species to let readers understand this aspect of the study. Both of these could be addressed by expanding Table 1. In general, readers would benefit greatly from much more informative figure captions.

- Sample size is reported now in the Materials and Methods section following the Introduction, Page 4, Line 129, 143-144.

- The body length is included now in Tables 2-3, Page 8 and Tables 4-5, Page 10. 

- The figure captions are largely changed across the manuscript. 

I think that the role of size as a design factor could be brought out more effectively overall. The authors note in the Introduction that “ Larger wings generate more lift than wings of smaller area or span and are associated with increased thrust production and stability control of large species when swimming. The relationships between S, A, and BL is different in life history stages that affect the swimming performance of cetaceans (lines 74 – 77).” But scaling principles aren’t employed to make baseline predictions about patterns of variation of various traits with respect to body size. This would make the study much more powerful. For example, the study finds that “absolute” parameters differ more cross the sample than do “relative” ones (which should be defined); surely this would be what one predicts? Based on scaling alone, one would predict that “relative” variables, which are ratios or angles, would tend not to vary with animal size. Because this idea isn’t developed in the Introduction, or brought up in the Discussion, readers are never able to appreciate the significance of any variation, or its absence, across the size sample.

- Thank you for this valuable suggestion, it helped a lot in revision of the manuscript. We hypothesized that, apart from the known scaling of the fin planform with the body length and mass of cetaceans, the dimensionless parameters of cross-sections are not related with the species and body length. The obtained results supported this Hypothesis (I) as well as Hypothesis II that constraints on variability of a generic design are associated with the primary function of the fin as a fixed or flapping hydrofoil, Page 3, Line 92-98. We hope that these changes help making the main message of the manuscript clearer.

There is an additional aspect of study design related to body size that requires further attention in the text, at the very least. All hydrodynamic performance simulations were carried out at two flow speeds, 2 and 8 m/s, which are categorized as cruising and burst speed. However, no biological rationale is provided for the selection of the speeds as cruising vs. burst, and most importantly, the analysis assumes that a given cruising or burst speed is appropriate for animals that vary many-fold in body size. This assumes that 2 m/s, or 8 m/s, is the same for a 75 kg harbor porpoise as for a 2 or 3 metric ton minke whale. This assumption seems wrong. Larger animals move more quickly, so a comparable speed would be one based on a certain number of body lengths per second or Strouhal number. The authors must explain their approach to this issue and carefully justify it. The rest of the study is difficult to interpret in advance of resolving this issue, because comparisons of results of the smallest and largest species at the same absolute speed does not make biological or physical sense.

- Our choice of two speeds of swimming, namely 2 and 8 m/sec, was made as on the empirical allometric relationships between swim speed and body mass for marine mammals (speed=0.78mass0.10, Watanabe et al 2011) as on the observed swimming speeds of cetaceans. Predicted and observed speeds of swimming were summarized in Table 1 in the “Hydrodynamic performance of the cross-sections of fins” subsection of the Materials and Methods, Page 7. Two speeds, 2 m/sec and 8 m/sec, chosen for our CFD testing of the fin cross-sections fell within a range of observed sustained and fast swimming speeds, respectively, Page 6-7, Line 215-222.

Also, if a primary goal of the study, as suggested by the first paragraph, is to provide insight into evolutionary pathways for divergence of a generic shape driven by the distinct roles of the dorsal fins and flukes in stability control and thrust production, it would be helpful to readers if the authors could propose hypotheses about how these should differ between the two sets of structures, and indicate how their analyses will test these specific hypotheses. 

- We added two hypotheses about a generic fin shape in the Introduction section and specified how it will be verified Page 3, Line 92-110.

There is some general language that describes differences in structure and function of dorsal fins and tail flukes, but this is not translated into specific predictions for hydrodynamic characteristics, the primary subject of the analysis. If dorsal fins are single fixed wings with stability as a primary function, and tail flukes are paired oscillating wings that function to provide control and for which increased lift and delayed stall are critical (please explain why), then what do you predict the simulations might discover as key differences between them? How should this be affected by body size and flow speed? 

- We assumed that the fins involved in stabilization/maneuverability control and thrust production could have the distinctive features of their cross-sectional geometry. To check it, we compared fins with their engineered analogs performing a similar function and optimized for a certain range of the operational conditions. Our approach was to compare the dorsal fin cross-sections with the foils used in vertical stabilizers and the tail flukes’ cross-sections with the foils used in the aerobatic wings. This approach allowed comparison of the geometry and hydrodynamics of fin sections and foils in terms of standard airfoil parameters and hydrodynamic coefficients. Delayed stall is critical for the tail fluke’s cross-sectional geometry as it allows a greater angle of attack variations without producing excessive minimum pressure peaks at the leading edge, that can result in flow separation and cavitation, Page 19, Line 545-549. 

- To check the influence of the body size and swimming speed we used the range of Re numbers based on dimensions of the cross-sections and selected swimming speeds 2 and 8 m/sec, Table 6, Page 15, Table 10, Page 16-17, Files S4,S5,S7,S8. 

There is one comparison (Fig S25) that is said to demonstrate critical differences between dorsal fins and flukes, but they come from different species (common dolphin fluke and Atlantic white-beaked dolphin dorsal fin), which seems highly problematic. Is this the only analysis that is relevant?

- The S25 Figure (now Fig 20) is based now on the data calculated for the dorsal fin and tail flukes of D. delphis, Page 18, Line 495-496. Besides, new figures are added to demonstrate the difference in the planform (Fig 18), cross-sectional geometry (Figs 7,9,11,19) and hydrodynamic characteristics of the fin cross-sections and the appropriate foils (Figs 12-17).

The other major arena in which the analysis is weak concerns the phylogenetic relationships among the species in the sample. The species included in this study are not independent data points. Five of the seven study species, D. delphis, L. acutus, L. albirostris, T. truncatus, and G. melas, are members of a single family, the Delphinidae. P. phocoena is a member of the sister taxon, family Phocoenidae. B. acutorostrata is a member of not only a distinct family, Balaenidae, this family is in a completely different part of the cetacean suborder, the Mysticeti or baleen, rather than Odontoceti or toothed whales. 

- This work is done on the sample taken from six Odontoceti species. The only one specimen of the Mysticeti, namely B. acutorostrata was added for the illustrative purpose.

So, the degree of evolutionary relationship between species pairs varies greatly. I don’t mean to say that the authors need to conduct a full, phylogenetically-informed analysis. But, particularly if they wish to cast the study in an evolutionary light, they must acknowledge the evolutionary relationships among the taxa. The study species differ in body size, ecology, and phylogeny. Attributing observed differences to a single factor is therefore impossible, and interpretations must proceed with caution.

- We tried to emphasize these differences in first paragraph of the Material and Methods: “Measurements of the body length and fins were taken from representatives of five genera of the family Delphinidae and one genus of the family Phocoenidae, these having different body length, external morphology and specialization (Fig 1).” – Line 112-114. 

Finally, any time simulation approaches are adopted, particularly for biological applications, it is valuable to provide the reader with validation of some kind. How do the results of the code employed by the authors, DesignFOIL™, compare to results of empirical studies of the same phenomena? 

- The link to the validation of the DesignFoil with the wind tunnel data is added to the manuscript, Page 6, Line 202-203.

What is lost by carrying out analysis in 2D? I have no problem with the choice of 2D analysis, but this should be justified to the reader, and the paper should provide readers with concrete understanding of the potential shortcomings or errors produced by the simulation approach. 

- This study is focused on the analysis in 2D to gain insight into the way of optimization of a natural wing to the specific function. It is indicated now in the Introduction, Line 95-108.

- This knowledge could be the key in further 3D flow simulations of the bio-inspired flapping foils with more realistic scenarios including the span-wise and chord-wise fin bending. 

No simulation is perfect, so we should always treat simulation in a manner that identifies the results in which we have greatest confidence, and those that are less reliable. The authors need to be clearer about the strengths but also weaknesses of their simulations. Some of the results provided from simulations (Suppementary figures S1-20) have some type of error bar, others (S21-25) have none. Is this due only to the number of specimens? If so, does this mean that no error estimates are included? How are we to interpret this – with no error estimates, it is difficult to understand how to say whether two species are similar or different.

- The error bar is absent only in the B. acutorostrata data, as it was the only one specimen. In all cases the error bar is presented and it can be too small in some cases due to the appropriate error values. 

The paper’s conclusions are largely reasonable, but in are not always supported by the data. One striking shortcoming is that although hydrodynamics is a/the central focus of the study, none of the paper’s main figures present any results of the hydrodynamics investigations; these are reserved solely for the Supplemental Information. This does not make sense to me. If the authors think that the most important results concern some of the CFD analyses, they should identify these and select or prepare figures that convey these results.

- There are twelve new figures and two tables illustrating the different aspects of the morphology and hydrodynamics of fins added to the main text now.

The article is presented in a mostly intelligible fashion and is largely written in standard English. However, there are numerous places where clarity and precision could be improved through editing by an expert or native speaker of English.

- Done.

The research meets all applicable standards for the ethics of experimentation and research integrity.

I do not believe that the article adheres to appropriate reporting guidelines and community standards for data availability. The authors state that “All relevant data are within the manuscript and its Supporting Information files”, but this does not appear to be true. The results are available in the manuscript and supporting information, but not the underlying data, and perhaps there is some confusion on the part of the authors concerning data availability. Community members could view plots of the output from simulations of flow around cross-sections of dorsal fins and flukes, for example, but neither the outlines of the hydrofoils nor the raw outputs of the simulations are included as data files.

- Hydrodynamic parameters calculated for the dorsal fin and tail flukes’ cross-sections located at the base and top of the fins are added in the Supplementary (S4-S5 and S7-S8 Files). The appropriate hydrofoil outlines are also added in the Supplementary (S3, S6 Files).

- The polar plots both for the dorsal fin and tail flukes’ cross-sections are added to the main text, Figs 12-17.

---

## [Decision Letter · Decision Letter 1]

28 May 2021

PONE-D-20-22457R1

Form, function, and divergence of a generic fin shape in small cetaceans

PLOS ONE

Dear Dr. Pavlov,

Thank you for submitting your manuscript to PLOS ONE. After careful consideration, we feel that it has merit but does not fully meet PLOS ONE’s publication criteria as it currently stands. Therefore, we invite you to submit a revised version of the manuscript that addresses the points raised during the review process.

We look forward to receiving your revised manuscript.

Kind regards,

Songhai Li, Ph.D.

Academic Editor

PLOS ONE

Journal Requirements:

Reviewers' comments:

Reviewer's Responses to Questions

**Comments to the Author**

1. If the authors have adequately addressed your comments raised in a previous round of review and you feel that this manuscript is now acceptable for publication, you may indicate that here to bypass the “Comments to the Author” section, enter your conflict of interest statement in the “Confidential to Editor” section, and submit your "Accept" recommendation.

Reviewer #1: (No Response)

Reviewer #3: All comments have been addressed

2. Is the manuscript technically sound, and do the data support the conclusions?

Reviewer #1: Partly

Reviewer #3: Yes

3. Has the statistical analysis been performed appropriately and rigorously? 

Reviewer #1: Yes

Reviewer #3: Yes

4. Have the authors made all data underlying the findings in their manuscript fully available?

Reviewer #1: Yes

Reviewer #3: Yes

5. Is the manuscript presented in an intelligible fashion and written in standard English?

Reviewer #1: Yes

Reviewer #3: Yes

6. Review Comments to the Author

Reviewer #1: The alterations and additions to the manuscript structure go to great lengths to address and satisfy the original comments made on the manuscript by all the reviewers. I find the current version of the manuscript to be quite much clearer in its presentation of its goals and results. My only comment regards the discussion of optimization in the dorsal fin.

A discussion of the variability of the dorsal fin morphology (lines 611-615 primarily) is indeed very relevant, and may even warrant expansion. I believe the general point of this section to be sound but the wording choice is poor or at least under elaborates on the phenomena being pointed out here which can't be fully resolved solely with the data of this manuscript.

More specifically I disagree with the choice to say that there’s a “high” demand in performance on the tail fluke but a “low demand” on the dorsal fin. All of the animals in this study are, broadly going to require the same function of their tail fluke (propulsive force) hence they converge on a similar morphology and the same performance would be optimized for by all animals (ideally). This appears to be what the authors interpret as "high performance demand". But differing life modes among different species can lead to differing functional demands and thus the exaggeration or elimination of a structure (such as what is observed in the dorsal fins). Some species may have a life mode with a “high” performance demand on the dorsal fin causing the exaggeration of AR while others may have a “low” demand for the function of that feature driving its elimination. The current phrasing instead suggests the dorsal fin plays such a small role in terms of performance that its morphology is irrelevant to these animals. Whether this is or is not the case cannot be determined by the limited data presented in this paper as different comparisons would need to be made, phylogenetic context would need to be incorporated, and other measures of performance would need to be included. If the intent is not what I have read it to be as presented above than the phrasing of this section needs to be revisited. If it is than at a minimum this statement needs to be softened but I would encourage the authors to dwell on the ideas discussed here and either expanding their discussion or presenting this as another opportunity for future work.

Reviewer #3: I am satisfied that the authors have adequately addressed the concerns raised in review. I think that from the perspective of comparative biology, the study remains weak, but readers will judge for themselves.

7. PLOS authors have the option to publish the peer review history of their article (what does this mean?). If published, this will include your full peer review and any attached files.

Reviewer #1: No

Reviewer #3: No

---

## [Author Response · Author response to Decision Letter 1]

2 Jul 2021

Dear Editor and Reviewers,

The authors are grateful for the valuable comment and suggestions of Reviewer #1 that resulted in the improvements of the Discussion Section. Please find our comments below.

Reviewer #1: The alterations and additions to the manuscript structure go to great lengths to address and satisfy the original comments made on the manuscript by all the reviewers. I find the current version of the manuscript to be quite much clearer in its presentation of its goals and results. My only comment regards the discussion of optimization in the dorsal fin.

A discussion of the variability of the dorsal fin morphology (lines 611-615 primarily) is indeed very relevant, and may even warrant expansion. I believe the general point of this section to be sound but the wording choice is poor or at least under elaborates on the phenomena being pointed out here which can't be fully resolved solely with the data of this manuscript.

More specifically I disagree with the choice to say that there’s a “high” demand in performance on the tail fluke but a “low demand” on the dorsal fin. All of the animals in this study are, broadly going to require the same function of their tail fluke (propulsive force) hence they converge on a similar morphology and the same performance would be optimized for by all animals (ideally). This appears to be what the authors interpret as "high performance demand". But differing life modes among different species can lead to differing functional demands and thus the exaggeration or elimination of a structure (such as what is observed in the dorsal fins). Some species may have a life mode with a “high” performance demand on the dorsal fin causing the exaggeration of AR while others may have a “low” demand for the function of that feature driving its elimination. 

- Both the “high demand” and “low demand” expressions are removed from the Discussion Section. The “degree of specialization in a primary function” is used instead.

The current phrasing instead suggests the dorsal fin plays such a small role in terms of performance that its morphology is irrelevant to these animals. Whether this is or is not the case cannot be determined by the limited data presented in this paper as different comparisons would need to be made, phylogenetic context would need to be incorporated, and other measures of performance would need to be included. If the intent is not what I have read it to be as presented above than the phrasing of this section needs to be revisited. If it is than at a minimum this statement needs to be softened but I would encourage the authors to dwell on the ideas discussed here and either expanding their discussion or presenting this as another opportunity for future work.

- We included the following paragraph in the final part of Discussion to address the Reviewer 1 comment: “The obtained results provide an insight into the evolutionary pathways of a generic fin shape driven by specialization in a primary function. Further studies of the developmental, genetic, and environmental drivers of fins variation would be the key in understanding the mechanisms of shaping the performance envelope of species having different habitat preference and feeding specialization.”, Page 21, Line 610-614. 

Reviewer #3: I am satisfied that the authors have adequately addressed the concerns raised in review. I think that from the perspective of comparative biology, the study remains weak, but readers will judge for themselves.

---

## [Editor Report · Decision Letter 2]

19 Jul 2021

Form, function, and divergence of a generic fin shape in small cetaceans

PONE-D-20-22457R2

Dear Dr. Pavlov,

We’re pleased to inform you that your manuscript has been judged scientifically suitable for publication and will be formally accepted for publication once it meets all outstanding technical requirements.

Kind regards,

Songhai Li, Ph.D.

Academic Editor

PLOS ONE
---

## [Editor Report · Acceptance letter]

30 Jul 2021

PONE-D-20-22457R2 

Form, function, and divergence of a generic fin shape in small cetaceans

Dear Dr. Pavlov:

I'm pleased to inform you that your manuscript has been deemed suitable for publication in PLOS ONE. Congratulations! Your manuscript is now with our production department. 

Kind regards, 

on behalf of

Dr. Songhai Li 

Academic Editor

PLOS ONE